# High-Resolution Traffic Sensing with Probe Autonomous Vehicles: A Data-Driven Approach

**DOI:** 10.3390/s21020464

**Published:** 2021-01-11

**Authors:** Wei Ma, Sean Qian

**Affiliations:** 1Department of Civil and Environmental Engineering, The Hong Kong Polytechnic University, Hong Kong, China; wei.w.ma@polyu.edu.hk; 2Department of Civil and Environmental Engineering, Carnegie Mellon University, Pittsburgh, PA 15213, USA; 3H. John Heinz III Heinz College, Carnegie Mellon University, Pittsburgh, PA 15213, USA

**Keywords:** autonomous vehicle, LiDAR, camera, state estimation, traffic sensing, data-driven, traffic flow, NGSIM

## Abstract

Recent decades have witnessed the breakthrough of autonomous vehicles (AVs), and the sensing capabilities of AVs have been dramatically improved. Various sensors installed on AVs will be collecting massive data and perceiving the surrounding traffic continuously. In fact, a fleet of AVs can serve as floating (or probe) sensors, which can be utilized to infer traffic information while cruising around the roadway networks. Unlike conventional traffic sensing methods relying on fixed location sensors or moving sensors that acquire only the information of their carrying vehicle, this paper leverages data from AVs carrying sensors for not only the information of the AVs, but also the characteristics of the surrounding traffic. A high-resolution data-driven traffic sensing framework is proposed, which estimates the fundamental traffic state characteristics, namely, flow, density and speed in high spatio-temporal resolutions and of each lane on a general road, and it is developed under different levels of AV perception capabilities and for any AV market penetration rate. Experimental results show that the proposed method achieves high accuracy even with a low AV market penetration rate. This study would help policymakers and private sectors (e.g., Waymo) to understand the values of massive data collected by AVs in traffic operation and management.

## 1. Introduction

As the combination of a wide spectrum of cutting-edge technologies, autonomous vehicles (AVs) are destined to fundamentally change and reform the whole mobility system [1]. AVs have great potentials in improving safety and mobility [2,3,4], reducing fuel consumption and emission [5,6], and redefining civil infrastructure systems, such as road networks [7,8,9], parking spaces [10,11,12], and public transit systems [13,14]. Over the past two decades, many advanced driver assistance systems (ADAS) (e.g., lane keeping, adaptive cruise control) have been deployed in various types of production vehicles. Currently, both traditional car manufacturers and high-tech companies are competing to lead full autonomy technologies. For example, Waymo’s AVs alone are driving 25,000 miles every day in 2018 [15], and there have been commercialized AVs operating in multiple cities by Uber [16].

Despite the rapid development of AVs technologies, there is still a long way to reach the full autonomy and to completely replace all conventional vehicles with AVs. We will witness a long period over which AVs and conventional vehicles co-exist on public roads. How to sense, model and manage the mixed transportation systems presents a great challenge to public agencies. To the best of our knowledge, most current studies view AVs as controllers and focus on modeling and managing the mixed traffic networks [17]. For example, novel system optimal (SO) and user equilibrium (UE) models are established to include AVs [18,19], coordinated intersections are proposed to improve the traffic throughput [20,21,22], vehicle platooning strategies are developed to reduce highway congestion [23,24], and AVs can also complement conventional vehicles to solve last-mile problems [25,26]. However, there is a lack of studies in traffic sensing methods for the mixed traffic networks.

In this paper, we advocate the great potentials of AVs as moving observers in high-resolution traffic sensing. We note that traffic sensing with AVs in this paper is different from perception of AVs [27]. The perception of AV is the key to the safe and reliable AVs, and it refers to the ability of AVs in collecting information and extracting relevant knowledge from the environment through various sensors [28], while traffic sensing with AVs refers to estimating the traffic conditions, such as flow, density and speed using the information perceived by AVs [29]. To be precise, traffic sensing with AVs is built on top of the perception technologies on AV, and in this paper, we will discuss the impact of different perception technologies on traffic sensing.

In fact, a fleet of autonomous vehicles (AVs) can serve as floating (or probe) sensors, detecting and tracking the surrounding traffic conditions to infer traffic information when cruising around the roadway network. Enabling traffic sensing with AVs is cost effective. The AVs equipped with various sensors and data analytics capabilities may be costly. While costly, those sensors and data are used primarily to detect and track adjacent objects to enable safe AV driving in the first place. Therefore, there is no additional overhead cost of these data collections for traffic sensing, since it is a secondary use.

High-resolution traffic sensing is central to traffic management and public policies. For instance, local municipalities would need information regarding how public space (e.g., curbs) is being utilized to set up optimal parking duration limits; metropolitan planning agencies would need various types of traffic/passenger information, including travel speed, traffic density and traffic flow by vehicle classifications, as well as pedestrians and cyclists. In addition, non-emergent and emergent incidents are reported by citizens through the 911 system, respectively. Automated traffic sensing, both historical and in real time, can complement those systems to enhance their timeliness, accuracy and accessibility. In general, accurate and ubiquitous information of infrastructure and usage patterns in public space is currently missing.

By leveraging the rich data collected through AVs, we are able to detect and track various objects in transportation networks. The objects include, but are not limited to, moving vehicles by vehicle classifications, parked vehicles, pedestrians, cyclists, signage in public space. When all those objects in high spatio-temporal resolutions are being continuously tracked, those data can be translated to useful traffic information for public policies and decision making. The three key features of traffic sensing based on autonomous vehicles sensors are: inexpensive, ubiquitous and reliable. Those data are collected by automotive manufacturers for guiding autonomous driving in the first place, which promises great scalability in this approach. With minimum additional efforts, the same data can be effectively translated into information useful for the community. For instance, how much time in public space at a particular location is utilized by different classifications of vehicles and by what travel modes, respectively? Can we effectively evaluate the accessibility, mobility and safety of the mobility networks? The sensing coverage will become ubiquitous in the near future, provided with an increasing market share of autonomous vehicles. Data acquired from individual autonomous vehicles can be compared, validated, corrected, consolidated, generalized and anonymized to retrieve the most reliable and ubiquitous traffic information. In addition, this paper for traffic sensing also implies the future possibility of interventions for effective and timely traffic management. It enables real-time traffic monitoring, potentially safer traffic operation, faster emergency response, and smarter infrastructure management.

The rest of this paper focuses on a critical problem to estimate the fundamental traffic state variables, namely, flow, density and speed, in high resolution, to demonstrate the sensing power of AVs. In addition to traffic sensing, there are many aspects and data in community sensing that could be explored in the near future. For example, perception of AVs can be used for monitoring urban forest health, air quality, street surface roughness and many other applications of municipal asset management [30,31,32,33].

Traffic state variables (e.g., flow, density and speed) play a key role in traffic operation and management. Over the past several decades, traffic state estimation (TSE) methods have been developed for not only stationary sensors (i.e., Eulerian data) but also moving observers (i.e., Lagrangian data) [34]. Stationary sensors, including loop detectors, cameras and radar, monitor the traffic conditions at a fixed location. Due to the high installation and maintenance cost, the stationary sensors are usually sparsely installed in the network, and hence the collected data are not sufficient for the practical traffic operation and management [35]. Data collected by moving observers (e.g., probe vehicles, ride-sourcing vehicles, unmanned aerial vehicles, mobile phones) have a better spatial coverage and hence it enables cost-effective TSE in large-scale networks [36]. Though the TSE method with moving observer dates back to 1954 [37], recent advances in communication and Internet of Things (IoT) technologies have catalyzed the development and deployment of various moving observers in the real world. Readers are referred to Seo et al. [29], Wang and Papageorgiou [38] for a comprehensive review of existing TSE models.

To highlight our contributions, we present studies that are closely related to this paper. The moving observers can be categorized into four types: originally defined moving observers, probe vehicles (PVs), unmanned aerial vehicles (UAVs) and AVs. Their characteristics and related TSE models are presented as follows:Originally defined moving observers. The moving observer method for TSE was originally proposed by Wardrop and Charlesworth [37]. The proposed method requires a probe vehicle to transverse along the road and count the number of slower vehicles overtaken by the probe vehicle and the number of faster vehicles which overtake the probe vehicle [39]. Though the setting of the originally defined moving observers is too ideal for practice, it enlightened us on the value of using Lagrangian data for TSE.PVs. The PVs refer to all the vehicles that can be geo-tracked, and it includes, but is not limited to, taxis, buses, trucks, connected vehicles, ride-sourcing vehicles [40]. The PV data have great advantages in estimating speed, while it hardly contains density/flow information. Studies have explored the sensing power of PVs [41]. PV data are usually used to complement stationary sensor data to enhance the traffic state estimation [42,43]. PVs with spacing measurement equipment can estimate traffic flow and speed simultaneously [44,45,46,47].UAVs. By flying over the roads and viewing from top-view perspectives, UAVs are able to monitor a segment of road or even the entire network [48,49,50]. UAVs have the advantage of better spatial coverage, while extra purchase of UAVs and the corresponding maintenance cost are required. Traffic sensing with UAV has been extensively studied in recent years, including vehicle identification algorithms [50,51,52], sensing frameworks [53,54], and UAV routing mechanisms [55,56].AVs. AVs can be viewed as probe vehicles equipped with more sensors and hence have better perception capabilities. Not only can the AV itself be geo-tracked, the vehicles surrounded by AVs can also be detected and tracked. AVs also share some similarities with UAVs because AVs can scan a continuous segment of road. We believe that AVs fall in between the PVs and UAVs, and hence existing TSE methods can hardly be applied to AVs. Furthermore, there are few studies on TSE with AVs. Moreover, Chen et al. [57] presents a cyber-physical system to model the traffic flow near AVs based on flow theory, while the TSE for the whole road is not studied. Recently, Uber ATG conduct an experiment to explore the possibility of TSE using AVs [58] but no rigorous quantitative analysis is provided.

The characteristics of TSE methods using different sensors are compared in Table 1. One can see that AVs have unique characteristics that are different from any other moving observers. Given the unique characteristics of AVs, there is a great need to study the AV-based TSE methods. However, as discussed above, this research area is still under-explored. In view of this, we develop a data-driven framework that estimates high-resolution traffic state variables, namely flow, density and speed using the massive data collected by AVs. The framework clearly defines the task of TSE with AVs involved and considers different perception levels of AVs. A two-step TSE method is proposed under a low AV market penetration rate. While this paper focuses on the road level traffic state estimation, the proposed approach could be further extended to the network-wide TSE, which is left for future research. The main contributions of this paper are summarized as follows:We firstly raise and clearly define the problem of TSE with multi-source data collected by AVs.We discuss the functionality and role of various AV sensors in traffic state estimation. The sensing power of AVs is categorized into three levels.We rigorously formulate the AV-based TSE problem. A two-step framework that leverages the sensing power of AVs to estimate high-resolution traffic state variables is developed. The first step directly translates the information observed by AVs to spatio-temporal traffic states and the second step employs data-driven methods to estimate the traffic states that are not observed by AVs. The proposed estimation methods are data-driven and consistent with the traffic flow theory.The next generation simulation (NGSIM) data are adopted to examine the accuracy and robustness of the proposed framework. Experimental results are compelling, satisfactory and interpretable. Sensitivity analysis regarding AV penetration rate, sensor configuration, and perception accuracy will also be studied.

Since TSE with AV data has not been explored in previous studies, this paper is probably the first attempt to rigorously tackle this problem. To this end, we first review the existing AV technologies that can contribute to traffic sensing, then we rigorously formulate the TSE problem. Finally, we propose and examine the solution framework. An overview of the paper structure is presented in Figure 1.

Section 2 discusses the sensing power of AVs. Section 3 rigorously formulates the high-resolution TSE framework with AVs, followed by a discussion of the solution algorithms in Section 4. In Section 5, numerical experiments are conducted with NGSIM data to demonstrate the effectiveness of the proposed framework. Lastly, conclusions are drawn in Section 6.

## 2. Sensing Power of Autonomous Vehicles

In order to prepare for the rigorous formulation of the AV-based TSE framework, we first discuss different levels of AV perception capabilities and how they associate with traffic sensing in this section. We discuss various sensors installed on AVs and their relation to traffic sensing. Analogous to the automation level definitions from the Society of Automotive Engineers (SAE), we define three sensing levels of AVs. Lastly, we discuss a conceptual data center for processing the sensing data.

### 2.1. Sensors

In this section, we discuss different types of sensors used for AV perception and their potential usage for traffic sensing. Sensors for perception that are mounted on AVs include, but are not limited to, camera, stereo vision camera, LiDAR, radar and sonar [28].

A camera can detect shapes and colors, so it is widely used for object detection (e.g., signals, pedestrians, vehicles and lane marks). Due to its low cost, multiple cameras can be mounted on a single AV. Theoretically, studies have shown that camera data can be used for object detection, tracking and traffic sensing [62,63]. In practice, camera image does not contain depth (distance) information, the localization of vehicles is challenging when using a single camera. On the modern AV prototypes, cameras are usually fused with stereo vision camera system or LiDAR to perceive the surrounding environments. In particular, the shape and color information obtained from camera are essential for object tracking [64,65]. Stereo vision camera refers to a device with two or more cameras horizontally mounted. Stereo vision camera is able to obtain the depth information of each pixel from the slightly different images taken by its cameras.

Light detection and ranging (LiDAR) uses the pulsed laser beam to measure the distance to the detected object. LiDAR can also obtain the 3D shape of the detected object. The LiDAR used on AVs is typically 360${}^{\circ }$, and the detection range varies from 30 to 150 m, depending on makers, detection algorithms and weather conditions. Both LiDAR and stereo vision camera can be used for vehicle detection and 3D mapping. The system latency (time delay for processing the retrieved data) of stereo vision camera is higher than LiDAR, though the price of stereo vision camera is much cheaper [27]. Theoretically, either the LiDAR or stereo vision camera can be used to build the full AV perception system, while currently, most AVs use LiDAR as the primary sensor.

There are two types of radar-mounted on AVs. The short-range radar (SRR) is typically used for blind spot detection, parking assist and collision warning. The range for SRR is around 20 m [66]. Similarly, sonar, with its limited detection range (3 to 5 m), is also frequently used for blind spot detection and parking assist. Neither of the two sensors are considered as appropriate sensors for traffic sensing. In contrast, the long-range radar (LRR), which is primarily used for adaptive cruise control, can be potentially used for traffic sensing. The range for LRR is around 150 m and it is dedicated to detecting the preceding vehicle in its current lane.

To conclude, Table 2 summarizes a list of sensors that can be potentially used for traffic sensing based on Van Brummelen et al. [27], Thakur [67].

### 2.2. Levels of Perception

In this section, we discuss how to categorize the sensing power of AVs with sensors listed in Table 2. The Society of Automotive Engineers (SAE) proposed a six-level classification criteria for autonomous vehicles [68]. L1 AVs can conduct adaptive cruise control (ACC), which is fulfilled by the long-range radar. From the perspective of traffic sensing, the L1 AV can always detect the location and speed of its preceding vehicle in the same lane. From L2 to L5, AVs gradually take control from human drivers. To achieve that, AVs need to continuously monitor the surrounding traffic conditions. From the perspective of traffic sensing, L2–L5 vehicles can detect or track the vehicles in their surrounding areas. Here, we emphasize the difference between vehicle detection and vehicle tracking. Detection refers to the localization of a vehicle when it appears in the detection area of an AV, and tracking means that AV can keep track of a vehicle, as long as it is within the detection area. To be precise, the task of detection does not require one to “memorize” the detected vehicles in each time frame, while tracking requires the AV to keep track of the detected vehicles when they are within the detection range. Tracking is technically much more challenging than the detection. As of today, the detection technology is fairly mature, while the tracking technology is still not ready for real-world applications [69]. The reason for the difference is that the detection/tracking is conducted frame by frame on AVs. If the AV processes 30 frame per-second, tracking requires one to accurately detect all the vehicles in each frame and match them correspondingly, while detection does not require one to match the vehicles in different frames. The matching is challenging because vehicles often block each other at times, and this makes it difficult for machines to decide whether the detected vehicle is the same vehicle detected in previous frames. From the perspective of traffic sensing, detection only provides the locations of each vehicle, but tracking can provide additional speed information.

Analogous to the SAE’s automation level definitions, we define three levels of sensing power for AVs, as presented in Figure 2.

The precise descriptions of the three perception levels are as follows.

$S_{1}$: The primary task for $S_{1}$ is to track the preceding vehicle within the same lane, and this is originally used for the ACC. However, the speed and location of the preceding vehicle are obtained for TSE.$S_{2}$: In addition to $S_{1}$, the primary task for $S_{2}$ is to detect and locate surrounding vehicles of an AV. Only vehicle counting at each time frame is obtained, and the speed information is not required in $S_{2}$.$S_{3}$: In addition to $S_{2}$, the primary task for $S_{3}$ is to track every uniquely identified vehicle in the detection area, hence the location and speed of each vehicle are monitored over time by AVs in $S_{3}$.

Based on the definition of AV perception levels, $S_{1}$ requires a LRR dedicated for preceding vehicles, $S_{2}$ requires a LiDAR/radar system, and $S_{3}$ requires a comprehensive fusion of camera, LiDAR, and radar. To be precise, Section 2.3 discusses how different sensors are combined to fulfill different levels of sensing power.

### 2.3. Detection Area of AVs

We now define the surrounding area (or detection area) of AVs, which is used throughout the paper. The detection area of AVs depends on the sensor configurations. Figure 3 presents two configurations of AV sensors. In the model of nuScenes, various sensors are mounted at different locations of an AV, while Waymo integrates most of the sensors on top of the vehicle. Depending on various sensor configurations on different AVs, the detection area of AVs can be different [70].

In this paper, we adopt a simplified representation of the AV detection area, as presented in Figure 4.

The detection area in Figure 4 consists of two components: $D_{1}$ and $D_{2}$. $D_{1}$ is used for detecting the preceding vehicle and fulfilled by the LRR; $D_{2}$ is for detecting all the surrounding vehicles, which is supported by the combination of LiDAR and cameras. We assume only $D_{1}$ is active in $S_{1}$, while both $D_{1}$ and $D_{2}$ are active in $S_{2}$ and $S_{3}$, as presented in Table 3. Within the detection range, we assume that the AV can measure the distance between itself and surrounding vehicles with a zero mean distance error, and the impact of the error will be quantified in the numerical experiments.

### 2.4. Centralized Data Communication, Collection, and Processing

In this paper, we assume that there is a centralized data service (data center) that receives all the information sent by AVs, as presented in Figure 5. Due to the bandwidth and latency restrictions, AVs do not send all the raw data to the data center. Instead, they only send the location and speed of the surrounding vehicles if applicable. The main task for the data center is to aggregate the information and remove the redundant information when the same vehicle is detected multiple times by different AVs in $S_{2}$ and $S_{3}$. This task can be done by checking and matching the location of the detected vehicles. For example, the vehicle with a green rectangle in Figure 5 is detected by two AVs, hence two duplicate data points are sent to the data center and the data center is able to identify and clean these duplicate data points. The localization accuracy is usually within the size of a standard vehicle, hence the accuracy for matching and cleaning is high [73]. In the numerical experiments, we will conduct a sensitivity analysis to evaluate the impact of different matching accuracies.

## 3. Formulation

Now, we are ready to rigorously formulate the traffic state estimation (TSE) framework with AVs. We first present the notations, and then the traffic states variables are defined. A two-step estimation method is proposed: the first step directly translates the information observed by AVs to spatio-temporal traffic states, and the second step employs data-driven methods to estimate the traffic states that are not observed by AVs.

### 3.1. Notations

All the notations will be introduced in context, and Table 4 provides a summary of the commonly used notations for reference.

### 3.2. Modeling Traffic States in Time-Space Region

We consider a highway with |L| lanes, where L={0,1,⋯,|L|−1}. The operator |·| is the counting measure for countable sets. For each lane l∈L, we denote $X_{l}$ as the set of longitudinal locations on lane *l*. In this paper, we treat each lane as a one-dimensional line. Without loss of generality, we set the starting point of $X_{l}$ to be 0, hence $X_{l}=\left[0,\mu \left(X_{l}\right)\right]$, where $\mu (X_{l})$ is the length of lane *l*. Throughout the paper, we denote operator μ(·) as the Lebesgue measure in either one or two dimensional Euclidean space, and it represents the length or area for one or two dimensional space. Note in this paper we assume the length of each lane is the same $\mu \left(X\right)=\mu \left(X_{l}\right),\forall l\in L$, while the proposed estimation method can be easily extended to accommodate different lane lengths. We further discretize the road $X_{l}$ to |S| equal distance road segments and each road segment is denoted by $X_{ls}$, where s∈S is the index of the road segment and S={0,1,⋯,|S|−1}. Hence, we have $X_{ls}=\left[s\Delta_{X},\left(s+1\right)\Delta_{X}\right]$ and $\Delta_{X}=\frac{\mu (X)}{\left|S\right|}$. The above road discretization is visualized in Figure 6.

We denote *i* as the index of a vehicle and *I* as the set of all vehicle indices. We further define the location $\left(x^{i}\left(t\right),l^{i}\left(t\right)\right)$, speed $v^{i}\left(t\right)$, and space headway $h^{i}\left(t\right)$ at a time point t∈T, where $x^{i}\left(t\right)$ is the longitudinal location of vehicle *i* at time *t*, $l^{i}\left(t\right)$ is the lane in which vehicle *i* is located at time *t*, and *T* is the set of all time points in the study period. We assume that each vehicle *i* only enters the highway once. If a vehicle enters the highway multiple times, the vehicle at each entrance will be treated as a different vehicle. To obtain the traffic states, we construct the distance $d_{l}^{i}$ and time $t_{l}^{i}$ and headway area $\alpha_{l}^{i}$ from vehicle location $\left(x^{i}\left(t\right),l^{i}\left(t\right)\right)$, speed $v^{i}\left(t\right)$, and headway $h^{i}\left(t\right)$ for a vehicle *i* based on Edie [74]. Throughout the paper, each vehicle is represented by a point, which is located at the center of each vehicle shape. When the vehicle is at the border of two lanes, function $l^{i}\left(t\right)$ will randomly assign the vehicle to either of the lanes.

Suppose ${\underline{t}}^{i}$ denotes the time point when the vehicle enters the highway and ${\bar{t}}^{i}$ denotes the time point when the vehicle exits the highway, we denote the distance traveled and time spent on lane *l* by vehicle *i* as $d_{ls}^{i}$ and $t_{lh}^{i}$, respectively. Mathematically, $d_{ls}^{i}$ and $t_{lh}^{i}$ are presented in Equation (Equation 1).
(1)$$\begin{array}{ccc} d_{ls}^{i} & = & \mu \left(\left\{x^{i}\left(t\right)\in X_{ls}|{\underline{t}}^{i}\leq t\leq {\bar{t}}^{i},l^{i}\left(t\right)=l\right\}\right)\\ t_{lh}^{i} & = & \mu \left(\left\{t\in T_{h}|{\underline{t}}^{i}\leq t\leq {\bar{t}}^{i},l^{i}\left(t\right)=l\right\}\right) \end{array}$$

We use the headway area $\alpha_{l}^{i}$ to represent the headway between vehicle *i* and its preceding vehicle on lane *l* in the time-space region, and it is represented by Equation (Equation 2).
(2)$$\begin{array}{c} \alpha_{l}^{i}=\left\{\left(t,x\right)|{\underline{t}}^{i}\leq t\leq {\bar{t}}^{i},l^{i}\left(t\right)=l,x^{i}\left(t\right)\leq x\leq x^{i}\left(t\right)+h^{i}\left(t\right)\right\} \end{array}$$

When we have the trajectories of all vehicles on the road, we can model the traffic states of each lane in a time-space region. Without loss of generality, we set the starting point of *t* to be zero, hence we have T=[0,μ(T)], where μ(T) is the length of the study period. We discretize the study period *T* to |H| equal time intervals, where H={0,1,⋯,|H|−1}. We denote $T_{h}$ as the set of time points for interval *h*, where h∈H. Therefore, we have $T_{h}=\left[h\Delta_{H},\left(h+1\right)\Delta_{H}\right]$, where $\Delta_{H}=\frac{\mu (T)}{\left|H\right|}$. In this paper, we use uniform discretization for $X_{l}$ and *T* to simplify the formulation, while the proposed estimation methods work for the arbitrary discretization scheme.

We use $A_{l}\left(h,s\right)$ to denote a cell in the time-space region for road segment $X_{ls}$ and time period $T_{h}$, as presented in Equation (Equation 3).
$$\begin{array}{ccc} A_{l}\left(h,s\right) & = & T_{h}⊗X_{ls}\\ \; & = & \mathrm{Polygon}[\left(h\Delta_{H},s\Delta_{X}\right),\left(\left(h+1\right)\Delta_{H},s\Delta_{X}\right),\\ \; & \; & \left(\left(h+1\right)\Delta_{H},\left(s+1\right)\Delta_{X}\right),\left(h\Delta_{H},\left(s+1\right)\Delta_{X}\right)]\\ \; & = & \left\{\left(t,x\right)|h\Delta_{H}\leq t\leq \left(h+1\right)\Delta_{H},s\Delta_{X}\leq x\leq \left(s+1\right)\Delta_{X}\right\} \end{array}$$

We denote the headway area of vehicle *i* in cell $A_{l}\left(h,s\right)$ by $a_{l}^{i}\left(h,s\right)$, as presented in Equation (Equation 3). The headway area can be thought of as the multiplication of the space headway and time headway in the time-space region.
(3)$$\begin{array}{c} a_{l}^{i}\left(h,s\right)=A_{l}\left(h,s\right)\cap \alpha_{l}^{i} \end{array}$$

**Example** **1**(Variable representation in time-space region). *In this example, we illustrate the variables defined in the time-space region. We consider a one-lane road and the lane index is l. Furthermore, $X_{l}$ is segmented into 6 road segments ($X_{l0},\cdots ,X_{l5}$), and T is segmented into 10 time intervals ($T_{0},\cdots ,T_{9}$), as presented in Figure 7. The cell $A_{l}\left(0,4\right)$ is the intersection of $X_{l4}$ and $T_{0}$, $A_{l}\left(8,1\right)$ is the intersection of $X_{l1}$ and $T_{8}$.*
*Each green line in the time-space region represents the trajectory of a vehicle. In Figure 7, we highlight the first (i=0), second (i=1) and the 8th (i=7) vehicle trajectory. The distance traveled by each vehicle is the same, hence $d_{l}^{i}=\mu \left(X_{l}\right)=\mu \left(X\right)$. We also highlight $t_{l}^{7}$ in Figure 7, which represents the time spent by each vehicle i=7 on lane l.*

*The headway area of vehicle i=1, denoted by $\alpha_{l}^{1}$, is represented by the green shaded area. The red shaded area, which represents $a_{l}^{1}\left(1,2\right)$, is the intersection of $\alpha_{l}^{1}$ and $A_{l}\left(1,2\right)$, based on Equation (Equation 3).*


According to Seo et al. [46], Edie [74], we compute the traffic states variables, namely flow ${\bar{q}}_{l}\left(h,s\right)$, density ${\bar{k}}_{l}\left(h,s\right)$ and speed ${\bar{v}}_{l}\left(h,s\right)$, for each road segment $X_{ls}$ and time period $T_{h}$, as presented in Equation (Equation 5).
(4)$$\begin{array}{ccc} {\bar{q}}_{l}\left(h,s\right) & = & \frac{\sum_{i\in I}d_{ls}^{i}}{\sum_{i\in I}\mu \left(a_{l}^{i}\left(h,s\right)\right)}\\ {\bar{k}}_{l}\left(h,s\right) & = & \frac{\sum_{i\in I}t_{lh}^{i}}{\sum_{i\in I}\mu \left(a_{l}^{i}\left(h,s\right)\right)}\\ {\bar{v}}_{l}\left(h,s\right) & = & \frac{{\bar{q}}_{l}\left(h,s\right)}{{\bar{k}}_{l}\left(h,s\right)}=\frac{\sum_{i\in I}d_{ls}^{i}}{\sum_{i\in I}t_{lh}^{i}} \end{array}$$

We treat the traffic states, (e.g., flow ${\bar{q}}_{l}\left(h,s\right)$, density ${\bar{k}}_{l}\left(h,s\right)$ and speed ${\bar{v}}_{l}\left(h,s\right)$) estimated from full samples of vehicles *I* as ground truth and unknown. In the following sections, we will develop a data-driven framework to estimate the traffic states from the partially observed traffic information obtained from autonomous vehicles under different levels of perception power.

### 3.3. Challenges in the High-Resolution TSE with AVs

As summarized in Table 1, TSE with AVs is a unique problem that has not been explored. In this section, we highlight the difficulties of this unique problem, and further motivate the proposed traffic sensing framework in the following sections.

As the vehicle trajectories in high-resolution time-space region are complicated, the information collected by the AVs is also highly complicated and fragmented. To illustrate, we consider a two-lane road with three vehicles (one AV, and two conventional vehicles A and B), as shown in Figure 8. The trajectory of the AV is represented by the blue line, and the shaded area is the detection area of the AV. The detection area of the AV is represented by the shaded areas with blue and grey color. Even when the AV is on the lane 0, it can still detect the vehicles on lane 1, thanks to the characteristics of LiDAR and cameras. Hence, the blue area means that the AV is on the current lane, and the grey area means that the AV is on the other lane.

As shown in Figure 8, both the AV and vehicle A change lanes during the trip. Vehicle A changes lane at the middle of time interval 1, and the AV change lanes at the beginning of time interval 3. One can see that vehicle A can be detected by the AV in the cell (1,2) on both lane 0 and lane 1, and vehicle B can be detected in the cell (3,3) and (3,4) on lane 1. For the cell (1,3) on lane 1, it is not straightforward to see whether vehicle A is detected or not, hence rigorous mathematical formulations should be developed to determine which cells are observable and which cells should be estimated. In real-world congested roads, vehicles might change lanes frequently, and hence their trajectories can be very complicated. From the example, we can see that, due to the complicated vehicle trajectories on the roads, the information collected by the AVs is non-uniformly distributed in the time-space region, and most of the existing TSE methods cannot be applied to such data. Therefore, solving the TSE with AV data calls for a new estimation framework.

### 3.4. Overview of the Traffic Sensing Framework

In this section, we present an overview of the traffic sensing framework with AVs. We assume a subset of vehicles are AVs, namely $I^{A}\subseteq I$, where $I^{A}$ denotes the index set of all AVs. The goal for the traffic sensing framework is to estimate the density and speed of each cell in the time-space region, using the information observed by AVs. Once the speed and density are estimated accurately, the traffic flow can be obtained by the conservation law [75].

The TSE methods can be categorized into two types: model-driven methods and data-driven methods. The model-driven methods rely on physical models such as traffic flow theory, while the data-driven methods automatically learn the relationship between different variables. In the case of AV-based TSE, the observed information is fragmented and lacking in certain patterns, so it is challenging to establish the physical model. In contrast, the data-driven methods can be built easily, thanks to the massive data collected by the AVs. Therefore, this paper focuses on the data-driven approach.

The proposed framework consists of two major parts: direct observation and data-driven estimation, as presented in Figure 9.

In the direct observation step, density and speed are observed directly through AVs. Since AVs are moving observers [37], traffic states can only be observed partially for a set of time intervals and road segments (i.e. cells) in a time-space region. Section 3.5 will rigorously determine the set of cells that can be directly observed by AVs and compute the direct observations from information obtained by AVs. We will discuss the direct observation with different levels of sensing power. The second part aims at filling up the unobserved information with data-driven estimation methods. The functions $\Psi_{l}$ and $\Phi_{l}$ are used to estimate the unobserved density and speed on lane *l*, respectively. Details will be presented in Section 3.6.

### 3.5. Direct Observation

In this section, we present to compute traffic states using the information that is directly observed by AVs under different levels of perception. We define $O_{l}^{j},j\in \left\{1,2\right\}$ as the set of time-space indices (h,s) of the cells that are observed by the AVs, and *j* is the index of the detection area. The set $O_{l}^{1}$ presents all (h,s) detected by $D_{1}$, and $O_{l}^{2}$ present all (h,s) detected by $D_{2}$. Details are presented in Appendix A. Now we formulate the traffic states that can be directly observed by AVs under different levels of perception. As a notation convention, we use ⋆ to represent the information that cannot be directly observed by AVs, and ${\tilde{k}}_{l},{\tilde{v}}_{l}$ denote the directly observed density and speed, respectively.

#### 3.5.1. $S_{1}$: Tracking the Preceding Vehicle

In the perception level $S_{1}$, an AV can only detect and track its preceding vehicle, and hence its detection area for density and speed is $O_{l}^{1}$. The observed density and speed can be represented in Equation (Equation 5).
(5)$$\begin{array}{ccc} {\tilde{k}}_{l}\left(h,s\right) & = & \left\{\begin{array}{cc} \frac{\sum_{i\in I^{A}}t_{lh}^{i}}{\sum_{i\in I^{A}}\mu \left(a_{l}^{i}\left(h,s\right)\right)} & \mathrm{if}\;\left(h,s\right)\in O_{l}^{1}\\ ⋆ & \mathrm{else} \end{array}\right.\\ {\tilde{v}}_{l}\left(h,s\right) & = & \left\{\begin{array}{cc} \frac{\sum_{i\in I^{A}}d_{ls}^{i}}{\sum_{i\in I^{A}}t_{lh}^{i}} & \mathrm{if}\;\left(h,s\right)\in O_{l}^{1}\\ ⋆ & \mathrm{else} \end{array}\right. \end{array}$$

Equation (Equation 5) is proven to be an accurate estimation of the traffic states [46]. We note that some AVs also have a LRR mounted to track the following vehicle behind the AVs, and this situation can be accommodated by replacing the set $I^{A}$ with $I^{A}\cup \mathrm{Following}\left(I^{A}\right)$ in Equation (Equation 5), where $\mathrm{Following}(I^{A})$ represents all the vehicles that follow AVs in $I^{A}$.

When the AV market penetration rate is low, $O_{l}^{1}$ only covers a small fraction of all cells in the time-space region, especially for multi-lane highways. In contrast, $O_{l}^{2}$ covers more cells than $O_{l}^{1}$. Practically, it implies that the LiDAR and cameras are the major sensors for traffic sensing with AVs.

#### 3.5.2. $S_{2}$: Locating Surrounding Vehicles

In the perception level $S_{2}$, both $D_{1}$ and $D_{2}$ are enabled by the LRR, LiDAR and cameras, while $D_{2}$ can only detect the location of surrounding vehicles. Hence, the density can be observed in both $D_{1}$ and $D_{2}$, and the speed is only observed in $D_{1}$. The estimation method for $D_{1}$ cannot be used for $D_{2}$, since the preceding vehicles of the detected vehicle might not be detected, hence $\alpha_{l}^{i}\left(h,s\right)$ cannot be estimated accurately. Instead, $D_{2}$ provides a snapshot of the traffic density at a certain time point, and we can compute the density of time interval *h* by taking the average of all snapshots, as presented in Equation (Equation 6).
(6)$$\begin{array}{ccc} {\tilde{k}}_{l}\left(h,s\right) & = & \left\{\begin{array}{cc} \frac{\sum_{i\in A}t_{lh}^{i}}{\sum_{i\in A}\mu \left(a_{l}^{i}\left(h,s\right)\right)} & \mathrm{if}\;\left(h,s\right)\in O_{l}^{1}\\ \frac{1}{\mu (T_{h}^{o}\left(l,s\right))}\int_{t\in T_{h}^{o}\left(l,s\right)}\frac{|I_{l}^{o}\left(t,s\right)|}{\Delta_{X}}dt & \mathrm{else}\;\mathrm{if}\;\left(h,s\right)\in O_{l}^{2}\\ ⋆ & \mathrm{else} \end{array}\right.\\ {\tilde{v}}_{l}\left(h,s\right) & = & \left\{\begin{array}{cc} \frac{\sum_{i\in A}d_{ls}^{i}}{\sum_{i\in A}t_{lh}^{i}} & \mathrm{if}\;\left(h,s\right)\in O_{l}^{1}\\ ⋆ & \mathrm{else} \end{array}\right. \end{array}$$
where $T_{h}^{o}\left(l,s\right)$ represents the set of time indices when $X_{ls}$ is covered by the $D_{2}$ in $T_{h}$, and $I_{l}^{o}\left(t,s\right)$ represents the set of vehicles detected by $D_{2}$ on $X_{ls}$ at time *t*. Detailed formulations are presented in Appendix A.

#### 3.5.3. $S_{3}$: Tracking Surrounding Vehicles

In the perception level $S_{3}$, both localization and tracking are enabled by the LRR, LiDAR and cameras. In addition to the information obtained by $S_{2}$, speed information of surrounding vehicles in $D_{2}$ is also available. Similar to the density estimation, we first computed the instantaneous speed of a cell at a certain time point by taking the harmonic mean of all detected vehicles, and then the average speed of a cell is computed by taking the average of all time points, as presented by Equation (Equation 7).
(7)$$\begin{array}{ccc} {\tilde{k}}_{l}\left(h,s\right) & = & \left\{\begin{array}{cc} \frac{\sum_{i\in A}t_{lh}^{i}}{\sum_{i\in A}\mu \left(a_{l}^{i}\left(h,s\right)\right)} & \mathrm{if}\;\left(h,s\right)\in O_{l}^{1}\\ \frac{1}{\mu (T_{h}^{o}\left(l,s\right))}\int_{t\in T_{h}^{o}\left(l,s\right)}\frac{|I_{l}^{o}\left(t,s\right)|}{\Delta_{X}}dt & \mathrm{else}\;\mathrm{if}\;\left(h,s\right)\in O_{l}^{2}\\ ⋆ & \mathrm{else} \end{array}\right.\\ {\tilde{v}}_{l}\left(h,s\right) & = & \left\{\begin{array}{cc} \frac{\sum_{i\in A}d_{ls}^{i}}{\sum_{i\in A}t_{lh}^{i}} & \mathrm{if}\;\left(h,s\right)\in O_{l}^{1}\\ \frac{1}{\mu (T_{h}^{o}\left(l,s\right))}\int_{t\in T_{h}^{o}\left(l,s\right)}\mathrm{hmean}\left(\left\{v^{i}\left(t\right)|x^{i}\left(t\right)\in X_{ls},i\in I\right\}\right)dt & \mathrm{else}\;\mathrm{if}\;\left(h,s\right)\in O_{l}^{2}\\ ⋆ & \mathrm{else} \end{array}\right. \end{array}$$
where hmean(·) represents the harmonic mean. Though $S_{3}$ provides the most speed information, the directly observed density is the same for $S_{2}$ and $S_{3}$. Overall, the sensing power of AV increases as more cells are directly observed from $S_{1}$ to $S_{3}$. In the following section, we will present to fill the ⋆ using data-driven methods.

### 3.6. Data-Driven Estimation Method

In this section, we propose a data-driven framework to estimate the unobserved density and speed in ${\tilde{k}}_{l}\left(h,s\right),{\tilde{v}}_{l}\left(h,s\right)$. To differentiate the density (speed) before and after the estimation, we use ${\hat{k}}_{l}\left(h,s\right)$ and ${\hat{v}}_{l}\left(h,s\right)$ to represent the estimated density and speed for time interval *h* road segment *s* and lane *l*, while ${\tilde{k}}_{l}\left(h,s\right),{\tilde{v}}_{l}\left(h,s\right)$ denote the density and speed before the estimation (i.e. after the direct observation). The method consists of two steps: (1) estimate the unobserved density ${\hat{k}}_{l}\left(h,s\right)$ given the observed density ${\tilde{k}}_{l}\left(h,s\right)$; (2) estimate the unobserved speed ${\hat{v}}_{l}\left(h,s\right)$ given that the density ${\hat{k}}_{l}\left(h,s\right)$ is fully known from estimation and speed ${\tilde{v}}_{l}\left(h,s\right)$ is partially known from direct observation.

We present the generalized form for estimating the unobserved density and speed in Equations (Equation 8) and (Equation 9), respectively.
(8)$$\begin{array}{ccc} {\hat{k}}_{l}\left(h,s\right) & = & \Psi_{l}\left(h,s,{\left\{{\tilde{k}}_{l^{\prime }}\right\}}_{l^{\prime }}\right) \end{array}$$
(9)$$\begin{array}{ccc} {\hat{v}}_{l}\left(h,s\right) & = & \Phi_{l}\left(h,s,{\left\{{\tilde{v}}_{l^{\prime }}\right\}}_{l^{\prime }},{\left\{{\hat{k}}_{l^{\prime }}\right\}}_{l^{\prime }}\right) \end{array}$$
where $\Psi_{l}$ is a generalized function that takes the observed density ${\left\{{\tilde{k}}_{l}\right\}}_{l}$ and time/space index h,s as input and outputs the estimated density. $\Phi_{l}$ is also a generalized function to estimate speed, while its inputs include the observed speed ${\left\{{\tilde{v}}_{l}\right\}}_{l}$, the estimated density ${\left\{{\hat{k}}_{l}\right\}}_{l}$, and the time/space index h,s. In this paper, we propose matrix completion-based methods for $\Psi_{l}$, and both matrix completion-based and regression-based methods for $\Phi_{l}$. The details are presented in the following subsections.

#### 3.6.1. Matrix Completion-Based Methods

The matrix completion-based model can be used to estimate either density or speed. We first assume that densities (speeds) in certain cells are directly observed by the AVs, as presented in Equation (Equation 10).
(10)$$\begin{array}{ccc} {\hat{k}}_{l}\left(h,s\right) & = & {\tilde{k}}_{l}\left(h,s\right),\forall \left(h,s\right)\in O_{l}^{k}\\ {\hat{v}}_{l}\left(h,s\right) & = & {\tilde{v}}_{l}\left(h,s\right),\forall \left(h,s\right)\in O_{l}^{v} \end{array}$$
where we denote $O_{l}^{k}$ and $O_{l}^{v}$ as the detection area for density and speed of lane *l* in time-space region with a certain sensing power. Precisely, for $S_{1}$, $O_{l}^{k}=O_{l}^{1},O_{l}^{v}=O_{l}^{1}$; for $S_{2}$, $O_{l}^{k}=O_{l}^{1}\cup O_{l}^{2},O_{l}^{v}=O_{l}^{1}$; and for $S_{3}$, $O_{l}^{k}=O_{l}^{1}\cup O_{l}^{2},O_{l}^{v}=O_{l}^{1}\cup O_{l}^{2}$.

For each lane *l*, the estimated density ${\hat{k}}_{l}$ (or speed ${\hat{v}}_{l}$) forms a matrix in the time-space region, and each row represents a road segment *s*, and each column represents a time interval *h*. Some entries ($\left(h,s\right)\notin O_{l}^{k}$ or $\left(h,s\right)\notin O_{l}^{v}$) in the density matrix (or speed matrix) are missing. To fill the missing entries, many standard matrix completion methods can be used. For example, the naive imputation (imputing with the average values across all time intervals or across all cells), k-nearest neighbor (k-NN) imputation [76], and the singular-value decomposition (SVD)-based SoftImpute algorithm [77]. In the numerical experiments, the above-mentioned methods will be benchmarked using real-world data.

#### 3.6.2. Regression-Based Methods

The speed data can also be estimated by a regression-based model, given that the density ${\hat{k}}_{l}\left(h,s\right)$ is fully estimated. We train a regression model $f_{l}$ to estimate the speed from densities for lane *l*, as presented in Equation (Equation 11).
(11)$$\begin{array}{cc} {\hat{v}}_{l}\left(h,s\right)= & f_{l}{\hat{k}}_{l^{\prime }}\left(h^{\prime },s^{\prime }\right):l-\delta_{l}\leq l^{\prime }\leq l+\delta_{l},h-\delta_{h}\leq h^{\prime }\leq h,\\ & s-\delta_{s}\leq s^{\prime }\leq s+\delta_{s},l^{\prime }\in L,s\in S,h\in H \end{array}$$
where $\delta_{l},\delta_{h},\delta_{s}$ represent the number of nearby lanes, time intervals and road segments considered in the regression model. The intuition behind the regression model is that the speed of a cell can be inferred by the densities of its neighboring cells. The choice of Equation (Equation 11) is inspired by the traffic flow theory (e.g., fundamental diagrams and car-following models) as the interactions between vehicles result in the road volume/speed. A specific example of $f_{l}$ is the fundamental diagram [78], which is formulated as ${\hat{v}}_{l}\left(h,s\right)=f_{l}\left({\hat{k}}_{l}\left(h,s\right)\right)$ by setting $\delta_{l}=\delta_{h}=\delta_{s}=0$.

In this paper, we adopt a simplified function $f_{l}\left(\cdot \right)$ presented in Figure 10. Suppose we want to estimate the speed for cell 1; there are 12 neighboring cells (including cell 1) considered as inputs. The regression methods adopted in this paper are Lasso [79] and random forests [80]. We also map the cells to the physical roads in time *t* and t−1, as presented in Figure 11. Figure 11 shows a three-lane road in time *t* and t−1, and the numbers marked for each road segment exactly match the cell number in Figure 10.

## 4. Solution Algorithms

In this section, we discuss some practical issues regarding the traffic sensing framework proposed in Section 3.

### 4.1. Computation of $a_{l}^{i}\left(h,s\right)$

To obtain the ground truth (Equation (Equation 4)) and the observed density (Equation (Equation 5)), $a_{l}^{i}\left(h,s\right)$, which denotes the headway area of vehicle *i* in cell $A_{l}\left(h,s\right)$ (Equation (Equation 3)), needs to be computed in the time-space region. Moreover, $a_{l}^{i}\left(h,s\right)$ is computed by intersecting $A_{l}\left(h,s\right)$ and $\alpha_{l}^{i}$, and $A_{l}\left(h,s\right)$ can be represented by a rectangle in the time-space region. The headway area for vehicle *i*$\alpha_{l}^{i}$ is usually banded [46], which can be approximated by a polygon. Therefore, $a_{l}^{i}\left(h,s\right)$ can also be represented by a polygon, and the interaction of a rectangle (which is also a special polygon) and a polygon can be conducted efficiently [81].

### 4.2. Sampling Rate

As discussed in Section 2.4, AVs send messages to the data center periodically. Let the sampling rate denote the message sending frequency, and we assume that all AVs have the same sampling rate. When the sampling rate is high, the data center can obtain the density and speed information in high temporal resolution, hence the traffic sensing can be accurate. On the other hand, the sampling rate is limited by the bandwidth and latency of the message transmission network. In the numerical experiments, sensitivity analysis will be conducted to study the impact of sampling rate.

### 4.3. Cross-Validation

In the data-driven method presented in Section 3.6, the cross-validation is conducted for model selection in both matrix completion-based and regression-based methods.

In the matrix completion-based model, we use cross-validation to select the maximal rank in the SoftImpute and the number of nearest neighbors in the k-NN imputation [82]. To perform the cross-validation for the matrix completion, we randomly hide 10% of the matrix entries and run the imputation methods on the rest of entries. Then, we measure the imputation accuracy by comparing the imputed values and the actual values on the 10% hidden entries.

In the regression-based model, 5-fold cross-validation is performed to select the optimal parameter settings for different regression methods, such as the weight of regularization term in Lasso, number of base estimators in random forests.

## 5. Numerical Experiments

In this section, we conduct the numerical experiments with NGSIM data to examine the proposed TSE framework. All the experiments below are conducted on a desktop with Intel Core i7-6700K CPU @ 4.00GHz × 8, 2133 MHz 2 × 16GB RAM, 500GB SSD, and the programming language is Python 3.6.8.

### 5.1. Data and Experiment Setups

We use next generation simulation (NGSIM) data to validate the proposed framework. NGSIM data contain high-resolution vehicle trajectory data on different roads [83]. Our experiments are conducted on I-80, US-101 and Lankershim Boulevard, and the overviews of the three roads are presented in Figure 12. NGSIM data are collected using a digital video camera, and its temporal resolution is 100ms. Details of the three roads can be found in Alexiadis et al. [83], He [84].

We assume that a random set of vehicles are AVs and the AVs can perceive the surrounding traffic conditions. Given the limited information collected by AVs, we estimate the traffic states using the proposed framework. We further compare the estimation results with the ground truth computed from the full vehicle trajectory data. The normalized root mean squared error (NRMSE), symmetric mean absolute percentage error (SMAPE1, SMAPE2) will be used to examine the estimation accuracy, as presented by Equation (Equation 12). SMAPE2 is considered as a robust version of SMAPE1 [86]. All three measurements are unitless.
(12)$$\begin{array}{ccc} \mathrm{NRMSE}(z,\hat{z}) & = & \sqrt{\frac{\sum_{\nu \in M}{\left(z_{\nu }-{\hat{z}}_{\nu }\right)}^{2}}{\sum_{\nu \in M}z_{\nu }^{2}}}\\ \mathrm{SMAPE}1(z,\hat{z}) & = & \frac{1}{\left|M\right|}\sum_{\nu \in M}\frac{|z_{\nu }-{\hat{z}}_{\nu }|}{z_{\nu }+{\hat{z}}_{\nu }}\\ \mathrm{SMAPE}2(z,\hat{z}) & = & \frac{\sum_{\nu \in M}\left|z_{\nu }-{\hat{z}}_{\nu }\right|}{\sum_{\nu \in M}\left(z_{\nu }+{\hat{z}}_{\nu }\right)} \end{array}$$
where *z* is the true vector, $\hat{z}$ is the estimated vector, ν is the index of the vector, and M is the set of indices in vector *z* and $\hat{z}$. When comparing two matrices, we flatten the matrices to vector and then conduct the comparison.

Here, we describe all the factors that affect the estimation results. The market penetration rate denotes the portion of AVs in the fleet. In the experiments, we assume that AVs are uniformly distributed in the fleet. The detection area $D_{1}$ is a ray fulfilled by LRR and $D_{2}$ is a circle fulfilled by LiDAR. We assume that LiDAR has a detection range (radius of the circle) and it might also oversee a vehicle with a certain probability (referred to as missing rate). The AVs can be at one level of perception, as discussed in Section 2. The sampling rate of data center can be different. In addition, different data-driven estimation methods are used to estimate the density and speed, as presented in Section 3.6. We define LR1 and LR2 as Lasso regressions, and RF1 and RF2 as random forests regressions. The number 1 means that only cells 1 to 4 are used as inputs, while the number 2 means that all 12 cells in Figure 10 are used as inputs. SI denotes the SoftImpute, KNN denotes the k-nearest neighbor imputation, and NI denotes the naive imputation by simply replacing missing entries with the mean of each column.

**Baseline setting**: the market penetration rate of AVs is 5%. The detection range of LRR is 150 m, and the detection range of LiDAR is 50 m with 5% missing rate. The level of perception is $S_{3}$, and the speed is detected without any noise. The sampling rate of data center is 1 Hz. SI is used to estimate density and LR2 is used to estimate speed. We set |H|=90 and |S|=60.

### 5.2. Basic Results

We first run the proposed estimation method with the baseline setting. The estimation method takes around 7 minutes to estimate all three roads, and the most time consuming part is the information aggregation in the data center (discussed in Section 2.4) and the computation of Equation (Equation 7). The estimation accuracy is computed by averaging the NRMSE, SMAPE1 and SMAPE2 through all lanes, and the results are presented in Table 5. In addition to the unitless measures, we also include the mean absolute error (MAE) in the table.

In general, the estimation method yields accurate estimation on highways (I-80 and US-101), while it underperforms on the complex arterial road (Lankershim Boulevard). Estimation accuracy of speed is always higher than that of density, which is because the density estimation requires every vehicle being sensed, while speed estimation only needs a small fraction of vehicles being sensed [87].

**Estimation accuracy on different lanes.** We then examine the performance of the proposed method on each lane separately, and the estimation accuracy of each lane is summarized in Table 6.

One can read from Table 6 that the proposed method performs similarly on most lanes. One interesting observation is that the proposed method performs well on Lane 6 on I-80, Lane 5 on US-101 and Lane 1 on Lankershim Blvd, and those lanes are merged with ramps. This implies that the proposed method has the potential to work well on merging intersections.

The proposed method performs differently on lanes that are near the edge of roads. For example, the proposed method yields the worst density estimation and the best speed estimation on lane 1 of I-80, which is an HOV lane. The vehicle headway is relatively large on the HOV lane, hence estimating density is more challenging given limited detection range of LiDAR. In contrast, speed on HOV lane is relatively stable, making the speed estimation easy. In addition, the estimation accuracy of the Lane 4 of Lankershim Blvd is low, as a result of the physical discontinuity of the lane.

One noteworthy point is that the estimation accuracy also depends on the traffic conditions. For example, traffic conditions on the HOV lane of I-80 tend to be free-flow and low-density, so the estimation accuracy is different from other lanes which tend to be dense and congested.

To visually inspect the estimation accuracy, we plot the true and estimated density and speed in time-space region for Lane 2 and Lane 4 of all three roads in Figure 13 and Figure 14. It can be seen that the estimated density and speed resemble the ground truth; even the congestion is discontinuous in the time-space region (see Lankershim Blvd in Figure 13). Again, the Lane 4 of Lankershim Blvd is physically discontinuous, hence a large block of entries are entirely missing in time-space region (see the third row of Figure 14), and the blocked missingness may affect the proposed methods and increase the estimation errors.

#### Effects of Densities on Speeds in the Regression-Based Models

We use the LR2 method to estimate the speed. After fitting the LR2 model, we look at the fitted regression coefficients and interpret the coefficients from the perspective of traffic flow theories. In particular, we select Lane 2 on US-101 and summarize the fitted coefficients in Table 7. The regression coefficients for other lanes and networks can be found in the supplementary material.

The R-square for Lane 2 on US-101 is 0.832, indicating that the regression model is fairly accurate. From Table 7, one can see that the Intercept is positive and it represents the free flow speed when the density is zero. Coefficients for x1 to x12 are all negative with high confidence, and this implies that higher density generally yields lower speed.

Recall Figure 10; suppose we want to estimate the speed for cell 1, we refer to cell 1∼4 as the surrounding cells in the current lane and cell 5∼12 as the surrounding cells in the nearby lanes. The coefficients of x1 to x4 are the most negative, indicating the densities of the surrounding cells in the current lane have the highest impact on the speed. The densities of surrounding cells in the nearby lanes also have negative impact on the speed but the magnitude is lower.

### 5.3. Comparing TSE Methods with Different Types of Probe Vehicles (PVs)

In this section, we compare the proposed method with other TSE methods using different types of PVs. Consistent with Table 1, we consider the following three types of PVs:**Conventional PVs**: Only speed can be estimated using the conventional PVs, and the estimation method is adopted from Yu et al. [61].**PVs with spacing measurement**: the TSE can be conducted and both speed and density can be estimated. The estimation method is adopted from Seo et al. [46].**AVs**: The TSE can be conducted and both speed and density can be estimated. The estimation method is proposed in this paper.

All three methods are implemented with baseline setting, and we make 5% of PVs conventional PVs, PVs with spacing measurement, or AVs. The estimation accuracy in terms of SMAPE1 is presented in Figure 15.

One can see that, by using more information collected by AVs, the proposed method outperforms the other TSE methods using different PVs for both speed and density estimation. This experiment further highlights the great potential of using AVs for TSE.

### 5.4. Comparing Different Algorithms

In this section, we examine different methods in estimating density and speed. Recall in Section 3.6 that the matrix completion-based methods can estimate both density and speed, while the regression-based methods can only estimate the speed. We run the proposed estimation method with different combinations of estimation methods for density and speed, and the rest of the settings are the same as the baseline setting. To be precise, three methods are used to estimate density: naive Imputation (NI), k-nearest neighbor imputation (KNN) and SoftImpute (SI). Seven methods will be used to estimate speed, and they are NI, KNN, SI, LR1, LR2, RF and RF2. We plot the heatmap of SMAPE1 for each road separately, as presented in Figure 16.

The speed estimation does not affect the density estimation, as the density estimation is conducted first. SI always outperforms KNN and NI for density estimation. Different combinations of algorithms perform differently on each road. We use A-B to denote the method that uses A for density estimation and B for speed estimation. NI-LR2 on I-80, SI-LR2 on US-101 and SI-RF on Lankershim Blvd outperform the rest of the methods in terms of speed estimation. Overall, the SI-LR2 generates accurate estimation for all three roads.

### 5.5. Impact of Sensing Power

We analyze the impact of sensing power of AVs on the estimation accuracy. Recalling Section 2.2 and Section 3.5, we consider three levels of perception for AVs. Based on Equations (Equation 5)–(Equation 7), more entries in the time-space region are directly observed when the perception level increases. We run the proposed estimation method with different perception levels and different methods for speed estimation. Other settings are the same as the baseline setting. The heatmap of MSAPE1 for each road is presented in Figure 17.

As shown in Figure 17, the proposed method performs the best on US-101 and the worst on Lankershim Blvd. With 5% market penetration rate, at least S2 is required for I-80 and US-101 to obtain an accurate traffic state estimation. Similarly, S3 is required for Lankershim Blvd to ensure the estimation quality. Later, we will discuss the impact of market penetration rate on the estimation accuracy under different perception levels.

The estimation accuracy improves for all speed estimation algorithms and all three roads when the perception level increases. Different speed estimation algorithms perform differently on different roads within the same perception level. For example, in S2, the imputation-based methods outperform the regression-based method on I-80 and US-101, while the Lasso regression outperforms the rest on Lankershim Blvd. In S3, all the density estimation methods perform similarly on I-80 and US-101, while the regression-based method significantly outperforms the imputation-based methods on Lankershim Blvd in terms of density estimation.

### 5.6. Impact of AV Market Ppenetration Rate

To examine the impact of AV market penetration rate, we run the proposed method with different market penetration rates ranging from 0.03 to 0.7, and the rest of the settings are the baseline settings. The experimental results are presented in Figure 18.

Generally, the estimation accuracy increases when the AV market penetration rate increases. Moreover, 5% penetration rate is a tipping point for an accurate estimation for I-80 and US-101, while Lankershim Blvd requires larger penetration rate. To further investigate the impact of market penetration rate under different levels of perception, we run the experiment with different penetration rate under three levels of perception, and the results are presented in Figure 19.

One can read that S2 and S3 yield the same density estimation, as the vehicle detection is enough for density estimation. Better speed estimation can be achieved on S3, since more vehicles are tracked and the speeds are measured. Again, Figure 19 indicates that at least S2 is required for I-80 and US-101 to obtain an accurate traffic state estimation, and S3 is required for Lankershim Blvd to ensure the estimation quality. For S1 and S2 in Lankershim Blvd, the estimation accuracy for speed reduces when the market penetration rate increases, probably due to the overfitting issue of the LR2 method.

We remark that AVs in S1 level are equivalent to connected vehicles with spacing measurements [46], and hence Figure 19 also presents a comparison between the proposed framework and the existing method. The results demonstrate that by using more information collected by AVs, the proposed framework outperforms the existing methods significantly when the market penetration rate is low.

In addition to the above findings, another interesting finding is that when the market penetration rate is low, the regression methods usually outperform the matrix completion-based methods, while the matrix completion-based methods outperform the regression-based methods when the market penetration rate is high.

### 5.7. Platooning

In the baseline setting, AVs are uniformly distributed in the fleet, while many studies suggest that a dedicated lane for platooning can further enhance mobility [88]. In this case, AVs are not uniformly distributed on the road. To simulate the dedicated lane, we view all vehicles on Lane 1 of I-80, Lane 1 of US-101, and Lane 1,2 of Lankershim Blvd as AVs, and all the vehicles on other lanes as conventional vehicles. To compare, we also set another scenario with the same number of vehicles, which are treated as AVs, uniformly distributed on roads. We run the proposed method on both scenarios with the rest of settings being the baseline setting, and the results are presented in Table 8.

As can be seen from Table 8, the distribution of AVs has a marginal impact on the estimation accuracy. The proposed method performs similarly on the scenarios of the dedicated lane and uniformly distribution for all three roads, which is probably because the detection range of LiDAR is large enough to cover the width of the roads.

### 5.8. Effects of Sensing Errors

As the object detection and tracking depend on the accuracy of sensors and algorithms, the ability of AVs in sensing varies. In this section, we study the effects of sensing errors on the estimation accuracy, and the sensing errors can be categorized into detection missing rate, speed detection noise, and distance measurement errors.

**Detection missing rate.** The AVs might overlook a certain vehicle during the detection, and we use the missing rate to denote the probability. We examine the impact of the missing rate by running the proposed estimation method with different missing rate ranging from 0.01 to 0.9, and the rest of the settings are the baseline settings. We plot the estimation accuracy for each road separately, as presented in Figure 20.

From Figure 20, one can read that the estimation error increases when the missing rate increases for all three roads. The density estimation is much more sensitive to the missing rate than the speed estimation. This is because overlooking vehicles have a significant impact on density estimation, while speed estimation only needs a small fraction of vehicles being observed.

**Noise level in speed detection.** We further look at the impact of noise in speed detection. We assume that the speed of a vehicle is detected with noise, and the noise level is denoted as ξ. If the true vehicle speed is ν, we sample $\bar{\xi }$ from the uniform distribution Unif(−ξ,ξ), and then the detected vehicle speed is assumed to be $\nu +\nu \bar{\xi }$. The reason that we define this noise is that the observation noise is usually proportional to the scale of the observation. We run the proposed estimation method by sweeping ξ from 0.0 to 0.4, and the rest of the settings are baseline settings. The estimation accuracy is presented in Figure 21.

Surprisingly, the proposed method is robust to the noise in speed detection, as the estimation errors remain stable when the speed noise level increases. One explanation for this is that the speed of each cell is computed by averaging the detected speeds from multiple vehicles, hence the detection noise is complemented and reduced based on the law of large numbers.

**Distance measurement errors.** When AVs detect or track certain vehicles, it measures the distance between itself and the detected/tracked vehicles in order to locate the vehicles. The distance measurement is either conducted by the sensors ( e.g., LiDAR) or computer vision algorithms, and hence the measurement might incur errors.

We categorize the distance measurement errors into two components: (1) frequency of the error happening, which is quantified by the percentage of the distance measurement that are associated with an error; (2) magnitude of the error, which is quantified by the number of cells that are offset from the true cell. For example, we assume that the distance measurement is 10% and 5 cells, that means 10% of the distance measures are associated with an error, and the detected location is at most 5 cells away from the true cell (the cell in which the detected vehicle is actually located).

To quantify the effects of the distance measurement errors, two experiments are conducted. Keeping the other settings the same as the baseline setting, we set the magnitude of the distance measurement as 5 cells, and then we vary the percentage of the errors from 5% to 50%. The results are presented in Figure 22. Similarly, we set the percentage of the errors to be 10% and vary the magnitude from 1 cell to 9 cells, and the results are presented in Figure 23.

Both figures indicate that the increasing frequency and magnitude of the distance measurement errors could reduce the estimation accuracy. The proposed framework is more robust on the magnitude of the measurement error, while it is more sensitive to the frequency of the measurement error.

### 5.9. Sensitivity Analysis

In this section, we examine the sensitivity of other important factors, (e.g., LiDAR detection range, sampling rate, and discretization size) in our experiments.

**LiDAR detection range.** The detection range of LiDAR varies in a wide range for different brands [89]. We run the proposed estimation method using different detection rage ranging from 10 m to 70 m, and the rest of the settings are the baseline settings. The estimation accuracy for each road is presented in Figure 24.

One can read that the estimation error reduces for both density and speed when the detection range increases. The gain in estimation accuracy becomes marginal when the detection range is large. For example, when the detection range exceeds 40 m on US-101, the improvement of the estimation accuracy is negligible. Another interesting observation is that, on Lankershim Blvd, even a 70-m detection range cannot yield a good density estimation with a 5% market penetration rate.

**Sampling rate.** Recall that the sampling rate denotes the frequency of the message (which contains the location/speed of itself and detected vehicles) sent to the data center, as discussed in Section 2.4. When the sampling rate is low, we conjecture that the data center received fewer messages, which increases the estimation error. To verify our conjecture, we run the proposed estimation method with different sampling rate ranging from 0.3 Hz to 10 Hz, and the rest of the settings are the base settings. The estimation accuracy on each road is further plotted in Figure 25.

The estimation accuracy increases when the sampling rate increases for all three roads, as expected. The density estimation is more sensitive to the sampling rate than the speed estimation. This is probably because the density changes dramatically in time-space region, while the speed is relatively stable.

**Different discretization sizes.** In this section, we demonstrate how the different discretization sizes affect the estimation accuracy. |H|=90 and |S|=60 in the baseline setting, and we change that to |H|=60 and |S|=40. The other settings remain the same, and the comparison results are presented in Table 9.

One can see that a bigger discretization size yields better estimation accuracy, because the speed and density change more stably on larger cells. This result also suggests that higher-granular TSE is more challenging and hence it requires more coverage of the observation data. A proper discretization size should be chosen based on the requirements for the estimation resolution and the available data coverage.

## 6. Conclusions

This paper proposes a high-resolution traffic sensing framework with probe autonomous vehicles (AVs). The framework leverages the perception power of AVs to estimate the fundamental traffic state variables, namely, flow, density and speed, and the underlying idea is to use AVs as moving observers to detect and track vehicles surrounded by AVs. We discuss the potential usage of each sensor mounted on AVs, and categorize the sensing power of AVs into three levels of perception. The powerful sensing capabilities of those probe AVs enable a data-driven traffic sensing framework which is then rigorously formulated. The proposed framework consists of two steps: (1) directly observation of the traffic states using AVs; (2) data-driven estimation of the unobserved traffic states. In the first step, we define the direct observations under different perception levels. The second step is done by estimating the unobserved density using matrix-completion methods, followed by the estimation of unobserved speed using either matrix completion methods or regression-based methods. The implementation details of the whole framework are further discussed.

The next generation simulation (NGSIM) data are adopted to examine the accuracy and robustness of the proposed framework. The proposed estimation framework is examined extensively on I-80, US-101 and Lankershim Boulevard. In general, the proposed framework estimates the traffic states accurately with a low AV market penetration rate. The speed estimation is always easier than density estimation, as expected. Results show that, with 5% AV market penetration rate, at least S2 is required for I-80 and US-101 to obtain an accurate traffic state estimation, while S3 is required for Lankershim Blvd to ensure the estimation quality. During the estimation of speed, all the coefficients in the Lasso regression are consistent with the fundamental diagrams. In addition, a sensitivity analysis regarding AV penetration rates, sensor configurations, speed detection noise and perception accuracy is conducted.

This study would help policymakers and private sectors (e.g Uber, Waymo and other AV manufacturers) understand the values of AVs in traffic operation and management, especially the values of massive data collected by AVs. Hopefully, new business models to commercializing the data [90] or collaborations between private sectors and public agencies can be established for smart communities. In the near future, we will examine the sensing capabilities of AVs at network level and extend the proposed traffic sensing framework to large-scale networks. We also plan to develop a traffic simulation environment to enable the comprehensive analysis of the proposed framework under different traffic conditions. Another interesting research direction is to investigate the privacy issue when AVs share the observed information with the data center.

## Figures and Tables

**Figure 1 sensors-21-00464-f001:**
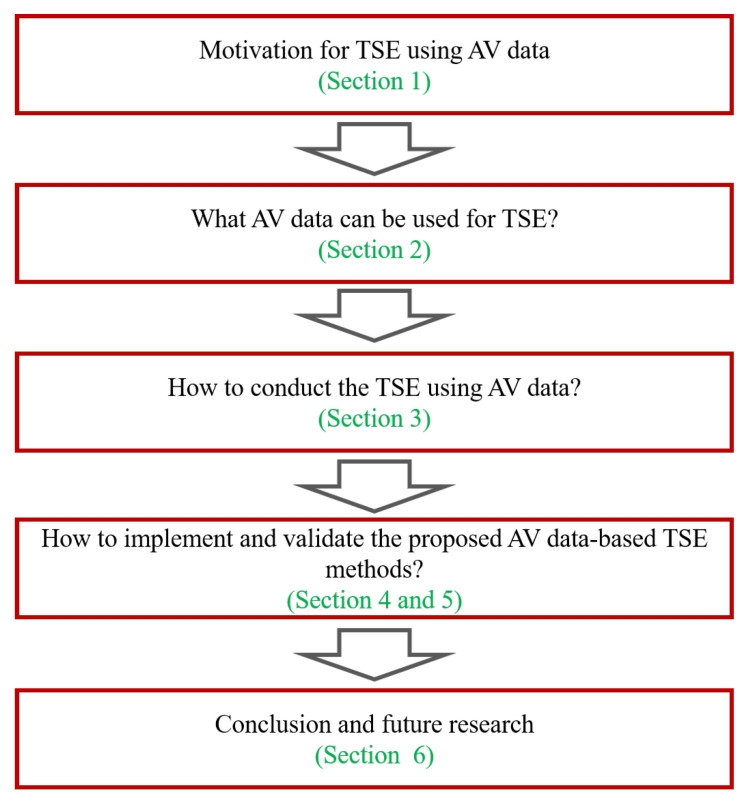
An overview of the paper structure.

**Figure 2 sensors-21-00464-f002:**
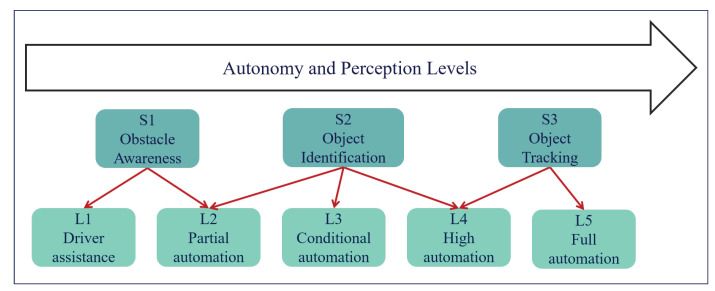
Overview of the perception levels.

**Figure 3 sensors-21-00464-f003:**
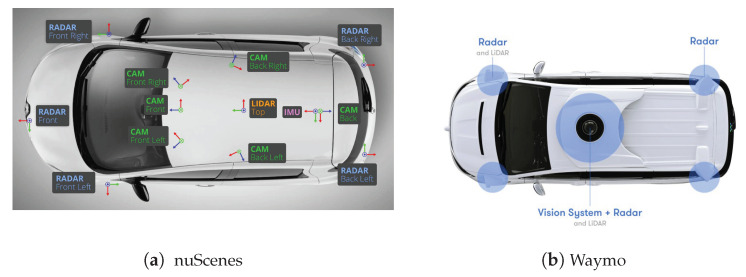
Two examples of sensor configurations, from nuScenes [71], Waymo Team [72]

**Figure 4 sensors-21-00464-f004:**
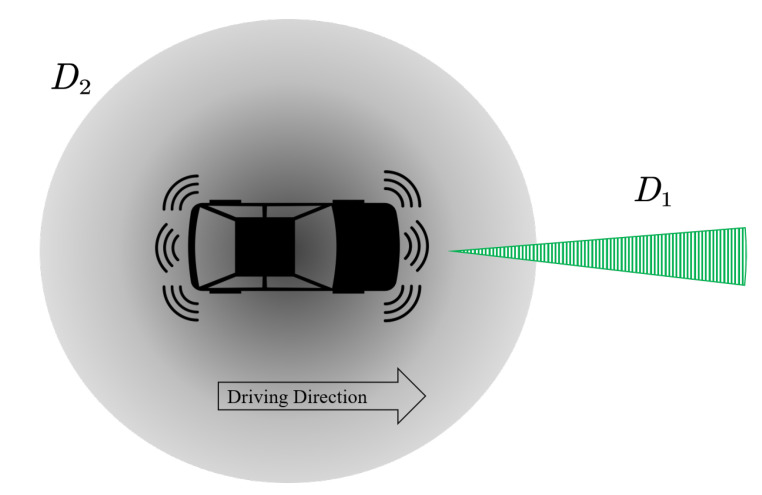
A simplified representation of AV detection area.

**Figure 5 sensors-21-00464-f005:**
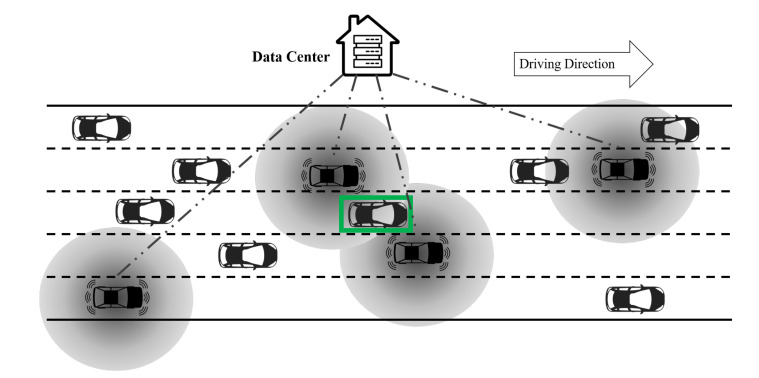
An illustration of the data center.

**Figure 6 sensors-21-00464-f006:**
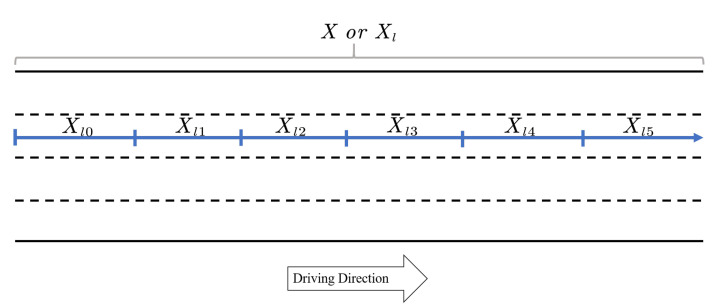
An illustration of the highway discretization.

**Figure 7 sensors-21-00464-f007:**
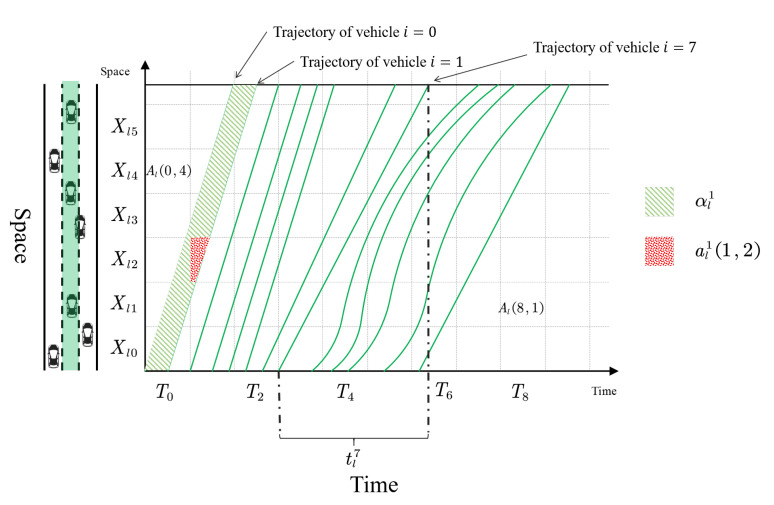
An example of variables in time-space region (the time-space region is associated with the green-colored lane, vehicles on the left subplot are for illustration purpose and do not exactly match with the trajectories on the left subplot).

**Figure 8 sensors-21-00464-f008:**
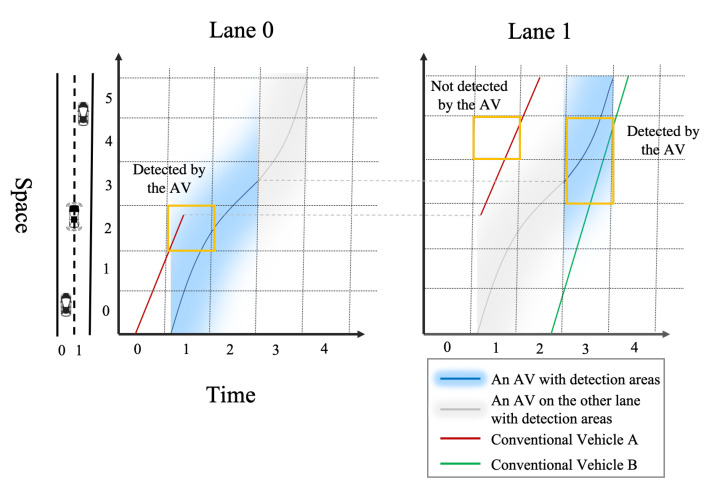
An illustrative example of the complicated and fragmented information collected by AVs (colored version available online).

**Figure 9 sensors-21-00464-f009:**
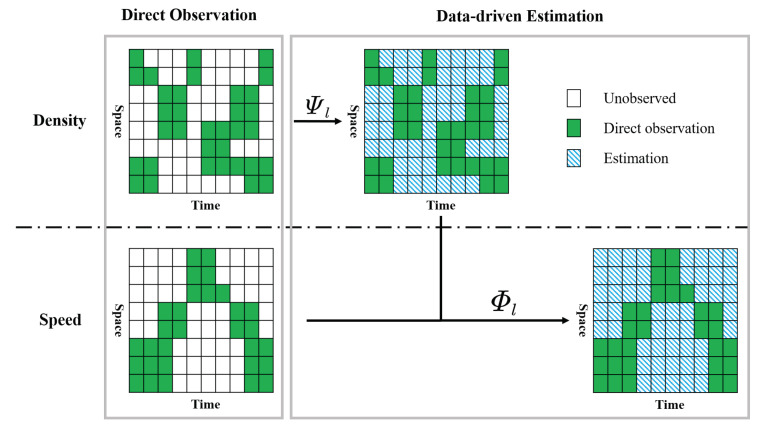
An overview of the traffic sensing framework for each one lane.

**Figure 10 sensors-21-00464-f010:**
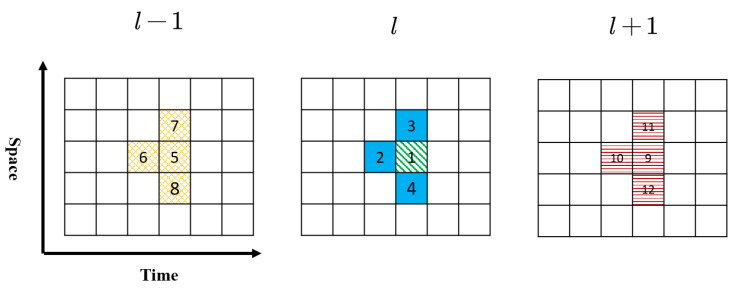
Cells in time-space region used for speed estimation.

**Figure 11 sensors-21-00464-f011:**
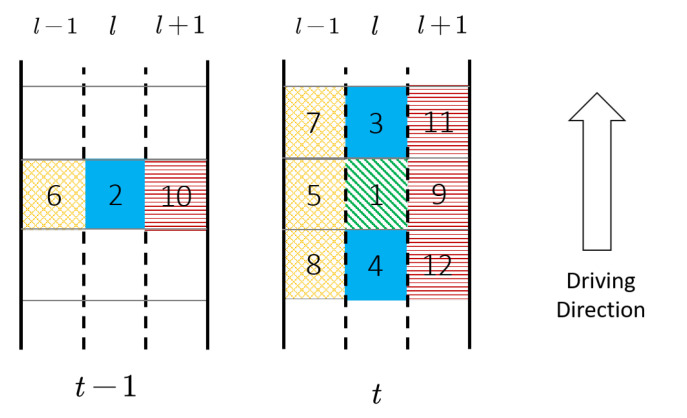
Cells in physical road used for speed estimation.

**Figure 12 sensors-21-00464-f012:**
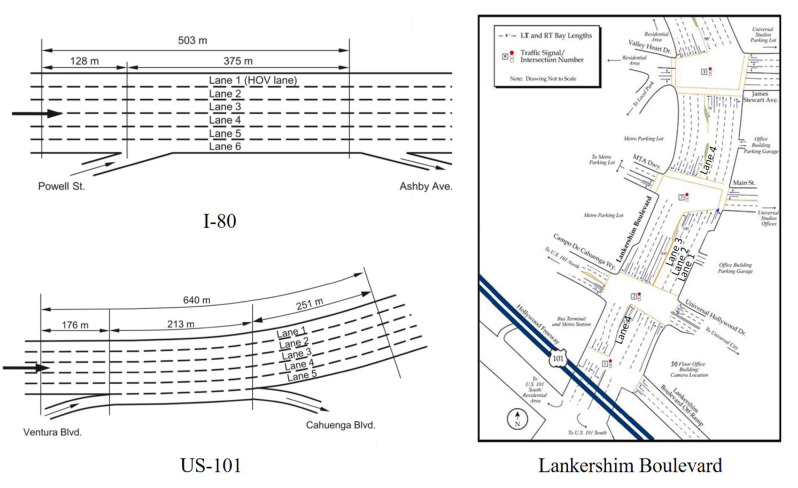
Overview of three networks(adapted from NGSIM website [83] and He [84], high-resolution figures are available at FHWA [85]).

**Figure 13 sensors-21-00464-f013:**
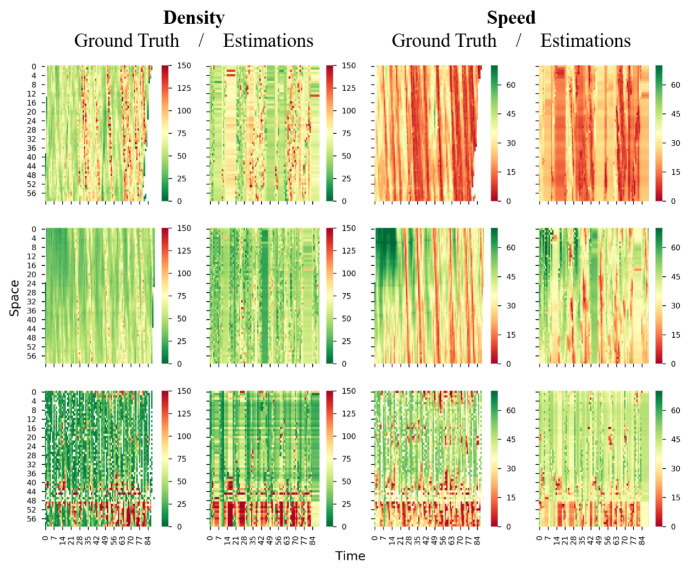
True and estimated density and speed for lane 2 (first row: I-80, second row: US-101. third row: Lankershim Blvd; first column: ground truth density, second column: estimated density, third column: ground truth speed, fourth column: estimated speed; density unit: veh/km, speed unit: km/h).

**Figure 14 sensors-21-00464-f014:**
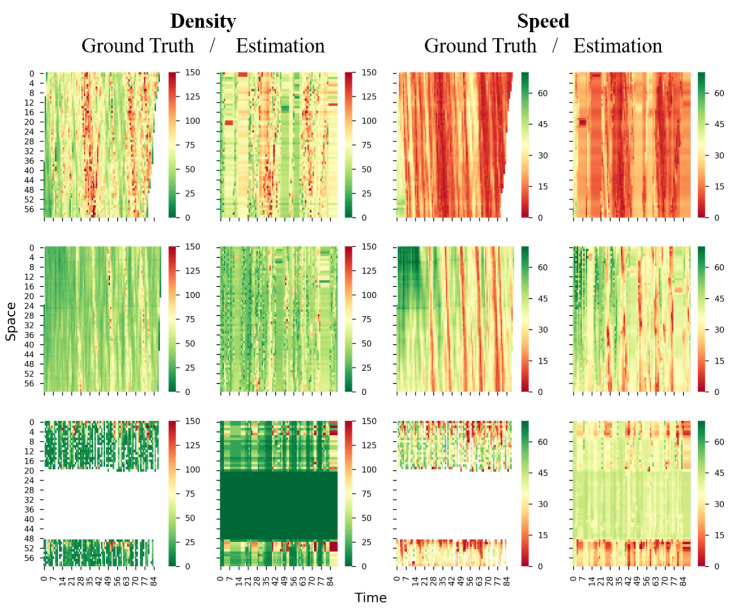
True and estimated density and speed for lane 4 (first row: I-80, second row: US-101. third row: Lankershim Blvd; first column: ground true density, second column: estimated density, third column: ground true speed, fourth column: estimated speed; density unit: veh/km, speed unit: km/h).

**Figure 15 sensors-21-00464-f015:**
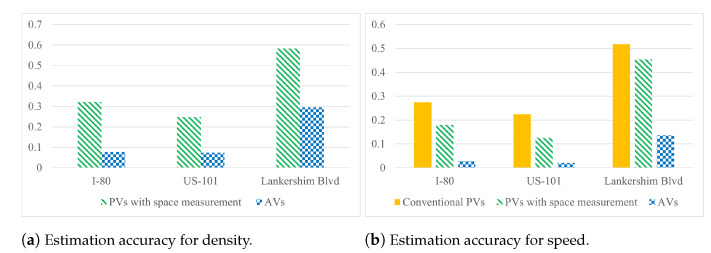
Average SMAPE1 for different types of PVs with baseline setting.

**Figure 16 sensors-21-00464-f016:**
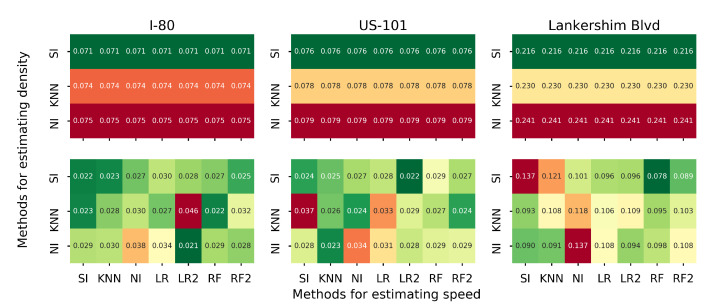
Average MSAPE1 for different estimation methods on each road (in terms of MSAPE1, first row: density, second row: speed).

**Figure 17 sensors-21-00464-f017:**
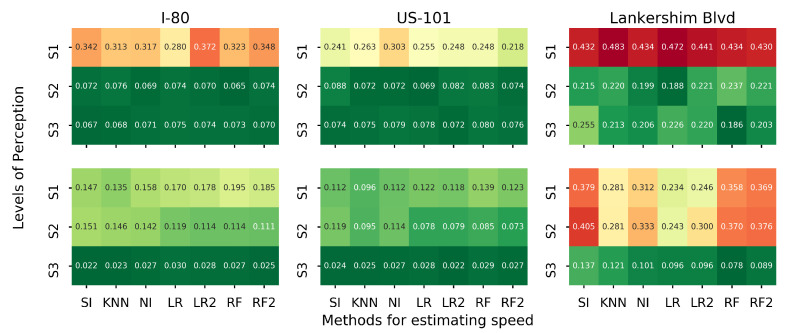
Estimation accuracy under three levels of perception (in terms of MSAPE1, first row: density, second row: speed).

**Figure 18 sensors-21-00464-f018:**
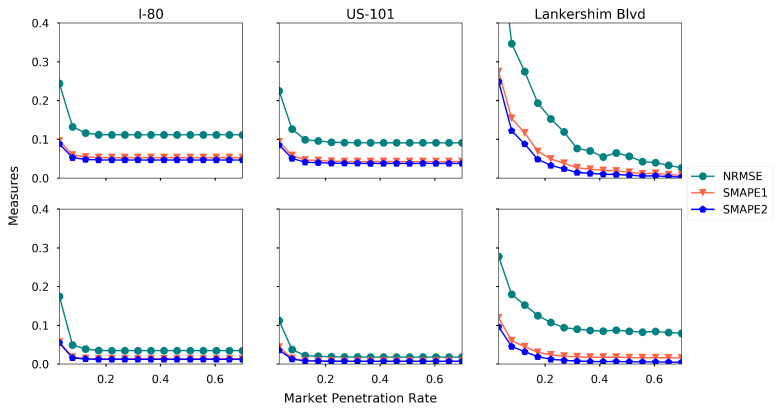
Estimation accuracy under different AV market penetration rate (first row: density, second row: speed).

**Figure 19 sensors-21-00464-f019:**
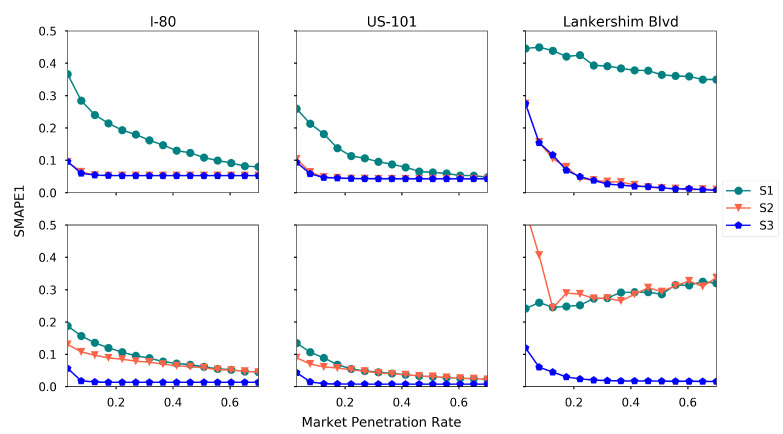
SMAPE1 under different market penetration rates and perception levels (first row: density, second row: speed).

**Figure 20 sensors-21-00464-f020:**
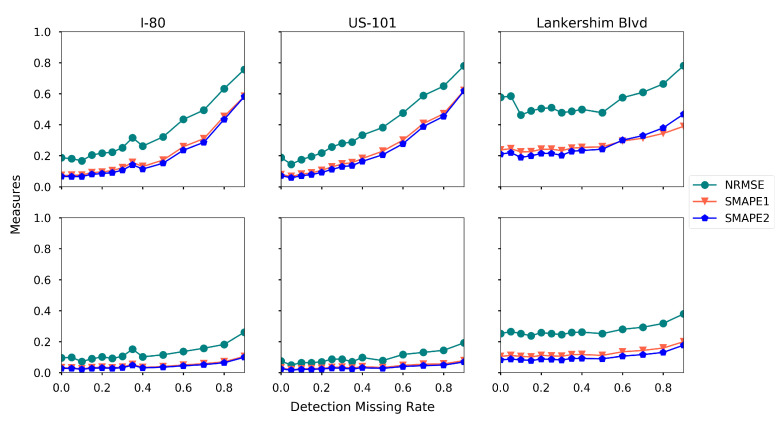
Estimation accuracy under different detection missing rate (first row: density, second row: speed).

**Figure 21 sensors-21-00464-f021:**
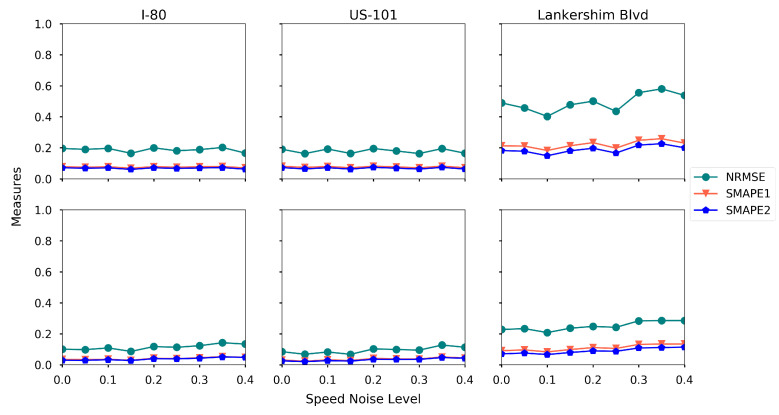
Estimation accuracy under different levels of speed noise (first row: density, second row: speed).

**Figure 22 sensors-21-00464-f022:**
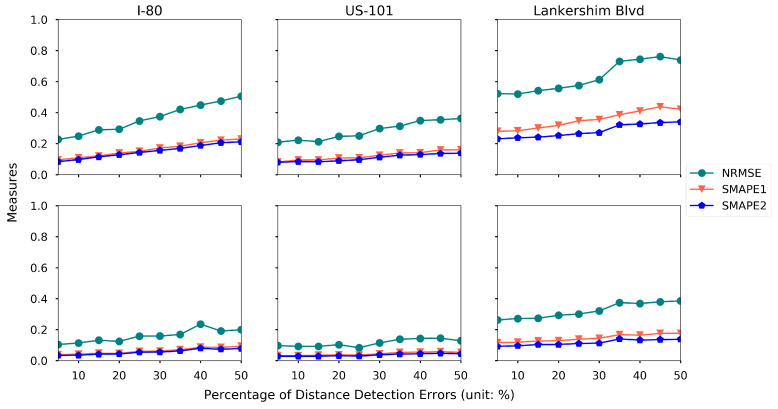
Estimation accuracy under different percentages of distance measurement errors (first row: density, second row: speed).

**Figure 23 sensors-21-00464-f023:**
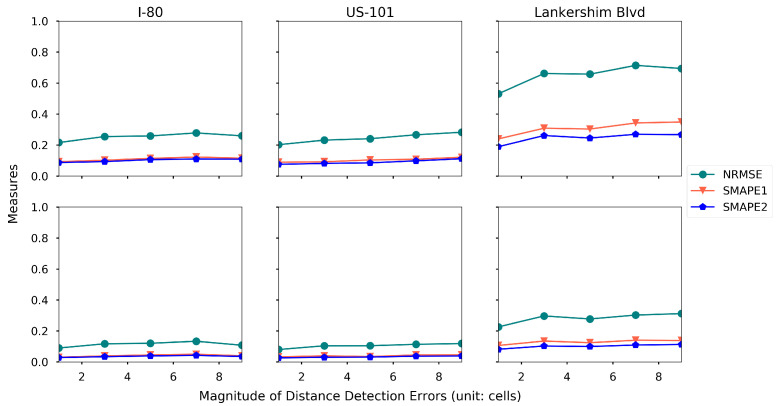
Estimation accuracy under different magnitude of distance measurement errors (first row: density, second row: speed).

**Figure 24 sensors-21-00464-f024:**
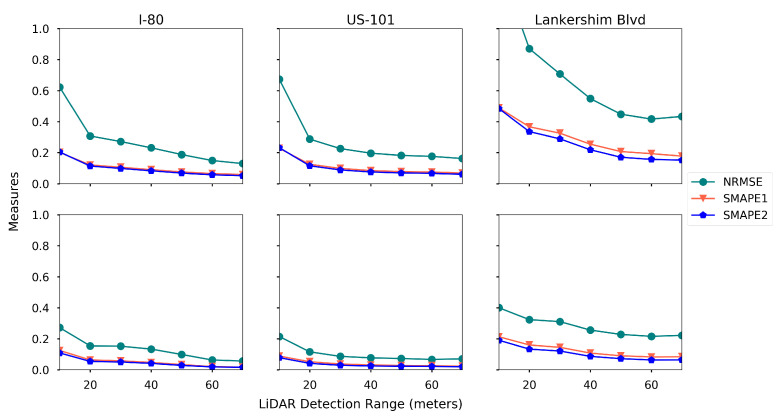
Estimation accuracy with different LiDAR detection range (first row: density, second row: speed).

**Figure 25 sensors-21-00464-f025:**
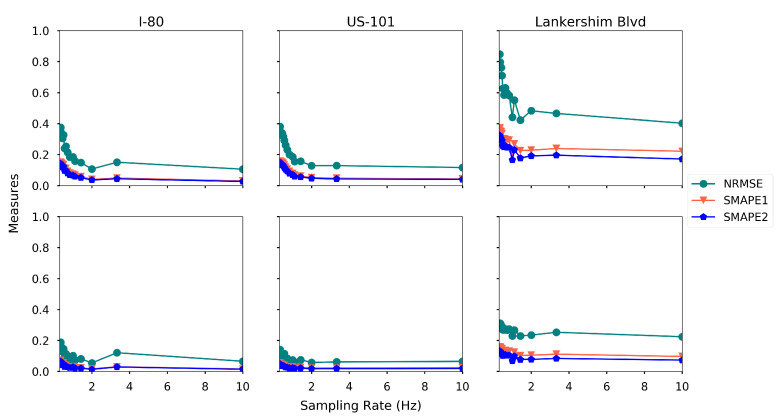
Estimation accuracy with different sampling rate (first row: density, second row: speed).

**Table 1 sensors-21-00464-t001:** Comparisons among TSE methods using different sensors.

	Stationary	Probe Vehicles (PVs)			UAVs
	Detectors	Conventional PVs	PVs with Spacing Measurement	AVs	
**Installed Sensors**	Loop detectors, cameras	GPS	GPS, LRR	GPS, LRR, LiDAR, cameras	GPS, cameras
**Raw Data Collected**	Vehicle counts over time	Trajectory of the PVs	Trajectory of the PVs and their first preceding vehicles	Trajectory of the AVs and their first preceding vehicles; **locations (or trajectories) of all the surrounding vehicles**	Birdviews of all vehicle locations
**Is TSE possible?**	Yes	No, only speed can be estimated	Yes	Yes	Yes
**Cost-effective**	No	Yes	Yes	Yes	No
**Easy to Deploy**	No	Yes	Yes	Yes	Yes
**Required Market Penetration Rate for TSE**	N/A	N/A	High	Low	Low
**Literature**	Wang and Papageorgiou [38], Thai and Bayen [59]	O’Keeffe et al. [41], Herring et al. [60], Yu et al. [61]	[44], Seo et al. [45], Seo et al. [46], Fountoulakis et al. [47]	This paper	Puri [48], Kanistras et al. [49], Ke et al. [50]

GPS: The Global Positioning System. LRR: Long Range Radar. LiDAR: Light Detection and Ranging.

**Table 2 sensors-21-00464-t002:** Sensors used for traffic sensing.

Sensors	Usage	Range
Camera	Surrounding vehicle detection/tracking, lane detection	20∼60 m
Stereo vision camera	Surrounding vehicle detection/tracking, 3D mapping	20∼60 m
LiDAR	Surrounding vehicle detection/tracking, 3D mapping	30∼150 m
Long-range radar	Preceding vehicle detection	150 m

**Table 3 sensors-21-00464-t003:** Summary of detection area and level of perceptions.

Sensing Power	Detection Area	Information Obtained
$S_{1}$	$D_{1}$	Speed/location of the preceding vehicle
$S_{2}$	$D_{1}$ and $D_{2}$	Speed/location of the preceding vehicle, location of surround vehicles
$S_{3}$	$D_{1}$ and $D_{2}$	Speed/location of the preceding vehicle and surrounding vehicles

**Table 4 sensors-21-00464-t004:** List of notations.

General Variables	
*l*	Index of a lane
*L*	The set of all lane indices *l*
*T*	The set of all time points in the study period
$T_{h}$	The set of all time points in time interval *h*
*x*	A longitudinal location along the road
$X_{l}$	The set of all longitudinal locations on lane *l*
⋆	Traffic state that is not directly observed by AVs
μ(·)	The Lebesgue measure for either one or two dimensional Euclidean space
|·|	The counting measure for the countable sets
**Variables in a Time-Space Region**	
*h*	Index of a time interval
*H*	The set of all indices *t* in the study period
*s*	Index of a longitudinal road segment
*S*	The set of all indices *s*
$X_{ls}$	The set of all longitudinal locations in road segment *s* and lane *l*
$A_{l}\left(h,s\right)$	A cell in time-space region for time interval *h* road segment *s* and lane *l*
${\bar{v}}_{l}\left(h,s\right)$	Average speed for time interval *h* road segment *s* and lane *l*
${\bar{q}}_{l}\left(h,s\right)$	Average traffic flow for time interval *h* road segment *s* and lane *l*
${\bar{k}}_{l}\left(h,s\right)$	Average density for time interval *h* road segment *s* and lane *l*
$a_{l}^{i}\left(h,s\right)$	The headway area of vehicle *i* in time-space region $A_{l}\left(h,s\right)$
**Variables Related to Vehicles**	
*i*	Index of a vehicle
*I*	The set of all vehicle indices *i*
$I_{l}\left(h,s\right)$	The set of all vehicles indices in time interval *h* road segment *s* and lane *l*
$v^{i}\left(t\right)$	Instantaneous speed of vehicle *i* at time *t*
$h^{i}\left(t\right)$	Instantaneous headway of vehicle *i* at time *t*
$x^{i}\left(t\right)$	Instantaneous longitudinal location of vehicle *i* at time *t*
$l^{i}\left(t\right)$	The lane in which vehicle *i* is located at time *t*
${\underline{t}}^{i}$	The time point when the vehicle *i* enters the road
${\bar{t}}^{i}$	The time point when the vehicle *i* exits the roads
$d_{ls}^{i}$	The distance traveled by vehicle *i* in road segment *s* on lane *l*
$t_{lh}^{i}$	The time spent in time interval *h* on lane *l* by vehicle *i*
**Variables Related to Autonomous Vehicles**	
*j*	Index of detection area
$D_{j}$	The detection area of an AV
$I^{A}$	The set of all autonomous vehicle indices
$O_{l}^{j}$	The set of time-space indices (h,s) such that $X_{ls}$ is covered by the AV detection area $D_{j}$ in time interval *h*
**Variables Related to the Sensing Framework**	
${\tilde{k}}_{l}\left(h,s\right)$	The directly observed density for time interval *h* road segment *s* and lane *l*
${\tilde{v}}_{l}\left(h,s\right)$	The directly observed speed for time interval *h* road segment *s* and lane *l*
${\hat{k}}_{l}\left(h,s\right)$	The estimated density and speed for time interval *h* road segment *s* and lane *l*
${\hat{v}}_{l}\left(h,s\right)$	The estimated speed for time interval *h* road segment *s* and lane *l*

**Table 5 sensors-21-00464-t005:** Estimation accuracy with basic setting (unit for NRMSE, SMAPR1, SMAPE2: %; unit for speed MAE: miles/hour; unit for density MAE: vehicles/miles).

Measures	Density				Speed			
	NRMSE	SMAPE1	SMAPE2	MAE	NRMSE	SMAPE1	SMAPE2	MAE
I-80	18.61	7.65	6.87	10.83	9.40	3.17	2.73	0.56
US-101	18.28	7.76	6.89	10.75	7.49	2.88	2.40	1.13
Lankershim	50.94	22.73	19.71	27.03	24.08	10.01	8.00	4.26

**Table 6 sensors-21-00464-t006:** Estimation accuracy on each lane with basic setting (unit: %).

Item	Lanes	I-80									
		1	2	3	4	5	6
Density	NRMSE	32.08	15.24	18.08	15.46	15.51	15.30			
	SMAPE1	13.43	6.21	7.17	6.02	6.45	6.61			
	SMAPE2	12.40	5.57	6.39	5.35	5.71	5.75			
Speed	NRMSE	6.82	9.68	9.17	10.99	9.99	9.73			
	SMAPE1	1.93	3.71	3.17	3.77	3.26	3.18			
	SMAPE2	1.94	3.01	2.73	3.17	2.83	2.73			
**Item**	**Lanes**	**US-101**					**Lankershim Blvd**			
		**1**	**2**	**3**	**4**	**5**	**1**	**2**	**3**	**4**
Density	NRMSE	17.90	17.95	18.47	18.09	18.96	45.49	41.30	43.75	73.19
	SMAPE1	7.56	7.62	7.60	7.75	8.27	20.92	21.00	21.37	27.63
	SMAPE2	6.75	6.79	6.83	6.79	7.28	17.53	16.59	16.95	27.73
Speed	NRMSE	8.76	7.85	6.86	7.18	6.76	23.94	21.33	25.26	25.78
	SMAPE1	3.62	3.05	2.65	2.56	2.50	10.22	8.77	10.97	10.08
	SMAPE2	2.87	2.48	2.22	2.19	2.21	7.71	6.77	8.49	9.02

**Table 7 sensors-21-00464-t007:** Coefficients of Lasso regression for Lane 2 on US-101 (x1 to x12 correspond to the cell number in Figure 10 or Figure 11).

Variables	Coefficients	Standard Error	t-Statistic	*p*-Value	2.5% Quantile	97.5% Quantile
Intercept	0.0778	0.000	254.909	0.000	0.077	0.078
x1	−0.4738	0.023	−20.676	0.000	−0.519	−0.429
x2	−0.2630	0.018	−14.326	0.000	−0.299	−0.227
x3	−0.4146	0.023	−17.700	0.000	−0.461	−0.369
x4	−0.4781	0.024	−20.346	0.000	−0.524	−0.432
x5	−0.2043	0.023	−9.030	0.000	−0.249	−0.160
x6	−0.0947	0.017	−5.726	0.000	−0.127	−0.062
x7	−0.1620	0.022	−7.352	0.000	−0.205	−0.119
x8	−0.2354	0.022	−10.562	0.000	−0.279	−0.192
x9	−0.1727	0.025	−6.790	0.000	−0.223	−0.123
x10	−0.1788	0.019	−9.477	0.000	−0.216	−0.142
x11	−0.1553	0.025	−6.198	0.000	−0.204	−0.106
x12	−0.2067	0.025	−8.256	0.000	−0.256	−0.158

**Table 8 sensors-21-00464-t008:** Estimation accuracy with AVs on dedicated lanes and uniformly distribution (A/B: A is for the uniformly distribution scenario, B is for the dedicated lane scenario, unit: %).

Measures	Density			Speed		
	NRMSE	SMAPE1	SMAPE2	NRMSE	SMAPE1	SMAPE2
I-80	31.71/31.65	12.45/12.47	11.86/11.85	19.78/19.84	8.08/7.81	6.92/6.95
US-101	25.93/25.91	10.18/10.16	9.62/9.61	15.13/15.15	6.05/6.05	4.90/4.91
Lankershim	66.67/66.71	31.17/31.58	28.89/28.70	38.99/37.60	20.03/19.74	16.00/15.67

**Table 9 sensors-21-00464-t009:** Estimation accuracy with AVs on baseline discretization and larger segmentation (A/B: A is for the baseline setting, B is for 40/60 discretization, unit: %).

Measures	Density			Speed		
	NRMSE	SMAPE1	SMAPE2	NRMSE	SMAPE1	SMAPE2
I-80	18.61/11.18	7.65/4.51	6.87/4.08	9.40/6.84	3.17/2.34	2.73/2.10
US-101	18.28/13.66	7.76/5.66	6.89/5.25	7.49/6.47	2.88/2.02	2.40/1.89
Lankershim	50.94/39.73	22.73/17.27	19.71/15.39	24.08/20.90	10.01/8.89	8.00/7.72

## Data Availability

The proposed traffic sensing framework is implemented in Python and open-sourced on Github (https://github.com/Lemma1/NGSIM-interface). Readers can reproduce all the experiments in Section 5. Additionally, the Github repository also contains some analysis that is omitted in Section 5.

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
