# Peer review of "High-Resolution Traffic Sensing with Probe Autonomous Vehicles: A Data-Driven Approach"

_sensors, 2021, doi:10.3390/s21020464_

Round 1

Reviewer 1 Report

This paper has performed a comprehensive modeling and examination to utilize discrete AVs as probes to estimate the traffic state. The results have been well compared and analyzed. The proposed method shows potential capability in estimating the traffic state. Even though the model and condition discussed in this paper are still relatively simple, it is reasonable to act as an guidance for further investigations.

The paper models the TSE on a high-way scenario and does not generalize it to town region or urban streets. The author should either verify the generality of this model capable for all scenarios, or constrain the model to some specific scenarios.

If multiple AVs sensed the same target vehicle, however, due to the inaccuracy of each AV sensing system (which will definitely happen if none of them are combined calibrated) there will be multiple “ghosts” of this single target vehicle. The reviewer suggests to make more complementary assumptions before digging into the modeling.

Equation (3). The reviewer is curious about what is the meaning of headway area, and how is it different from headway? The equation is clear but the meaning is not well stated.

Figure 7. Does the left subplot (vehicles on road) reflect the correct initial space condition when time = 0? Are the vehicle number i corresponds to this subplot? It seems this plot is confusing the reader since there are only 3 vehicles in the center lane.

Figure 8. Please make legend to show which car is AV and which is not. Moreover, the reviewer suggests the author to draw a few space schemes (vehicles on road) at various timing to make a better illustration of how each vehicle shall move at different time. Also note which car is A and which is B.

Section 5.3. The author compares the proposed method with two others. Can the author clarify if the author re-produces the other results, or directly copies the results stated in the references?

The nomenclatures of this paper should be provided since there are lots of author defined parameters. They are hard for the readers to follow.

Reviewer 2 Report

The subject of the article is very interesting and innovative. After reading this article, I report the following issues.

- line 27: "... to lead the full autonomy (L4 - 5) technologies." - it would be good to explain briefly levels L4 and L5, because the explanation is only on p. 6, lines 221 and the following lines.

- Figure 1.: "Section 1" is missing from the schematic. This is probably because "Introduction" was omitted as "Section 1". It is intuitive, however, for the formal order, it is proposed to supplement the diagram in Fig. 1 with "Section 1" as, for example, "Justification of the conducted research" or "Justification of the subject of AVs".

- Figure 1.: A similar remark to the previous one, ie "Section 6" is missing from the schema. It is recommended to supplement this lack - the more that in the paragraph to Fig.1 there is a text referring to "Section 6." - line 176.

- Figure2.: It is proposed to supplement the diagram with elements showing functional relationships between the relevant three perception levels S1, S2, S3 and the autonomous vehicles levels L1, L2, ..., L5. Maybe with the use of connecting arrows?

- Figure 10 and Figure 11: To facilitate the identification of cells on Fig. 10 for each lane (L-1, L, L + 1) in conjunction with Fig. 11 for each time (t-1, t) it is proposed to use 3 different colors or blue in 3 shades separately for each lane. Then it will be easier to find cells from Fig. 10 in Fig. 11.

- Figure 12.: "Lankershim Boulevard" - roads description illegible.

- Figures 13 and 14: It would be much better if the descriptions of "density" and "speed" were placed on the charts at the appropriate OX and OY axes.
